# On the damage tolerance of 3-D printed Mg-Ti interpenetrating-phase composites with bioinspired architectures

Mingyang Zhang [1,2,7], Ning Zhao[3,7], Qin Yu [4], Zengqian Liu [1,2 ✉], Ruitao Qu[1,5], Jian Zhang[1], Shujun Li[1,2], Dechun Ren[1,2], Filippo Berto[6], Zhefeng Zhang [1,2 ✉] & Robert O. Ritchie [4 ✉]

Bioinspired architectures are effective in enhancing the mechanical properties of materials, yet are difficult to construct in metallic systems. The structure-property relationships of bioinspired metallic composites also remain unclear. Here, Mg-Ti composites were fabricated by pressureless infiltrating pure Mg melt into three-dimensional (3-D) printed Ti-6Al-4V scaffolds. The result was composite materials where the constituents are continuous, mutually interpenetrated in 3-D space and exhibit specific spatial arrangements with bioinspired brick-and-mortar, Bouligand, and crossed-lamellar architectures. These architectures promote effective stress transfer, delocalize damage and arrest cracking, thereby bestowing improved strength and ductility than composites with discrete reinforcements. Additionally, they activate a series of extrinsic toughening mechanisms, including crack deflection/twist and uncracked-ligament bridging, which enable crack-tip shielding from the applied stress and lead to "Γ"-shaped rising fracture resistance R-curves. Quantitative relationships were established for the stiffness and strengths of the composites by adapting classical laminate theory to incorporate their architectural characteristics.

[1] Shi-Changxu Innovation Center for Advanced Materials, Institute of Metal Research, Chinese Academy of Sciences, 110016 Shenyang, China. [2] School of Materials Science and Engineering, University of Science and Technology of China, 230026 Hefei, China. [3] School of Materials Science and Engineering, Lanzhou University of Technology, 730050 Lanzhou, China. [4] Department of Materials Science and Engineering, University of California Berkeley, Berkeley, CA 94720, USA. [5] State Key Laboratory of Solidification Processing, School of Materials Science and Engineering, Northwestern Polytechnical University, 710072 Xi'an, China. [6] Department of Mechanical and Industrial Engineering, Norwegian University of Science and Technology, Richard Birkelands vei 2B, Trondheim 7034, Norway. [7] These authors contributed equally: Mingyang Zhang, Ning Zhao. ✉email: zengqianliu@imr.ac.cn; zhfzhang@imr.ac.cn; roritchie@lbl.gov

Strength and fracture toughness are two vital mechanical properties for structural materials, yet they often display a mutually exclusive relationship with each other. By taking lessons from natural biological materials, the bioinspired design of structural architectures has become an effective approach toward improved performance in man-made materials; in particular, it offers new opportunities for achieving synergetic enhancement in both strength and fracture toughness[1–3]. In this respect, different types of bioinspired architectures have been constructed in polymers and polymer-containing composite systems owing to their ease of processing[4–10]. Specifically, the easy additive manufacturing of polymers, e.g., by direct writing, makes them ideal prototypes for exploring the structure-property relationships for both biological and bioinspired materials[7–10]. Nevertheless, the structural applications of these materials are often limited by their low strengths and poor thermal resistance. The applicability of the structure-property relationships derived from polymeric materials to other material systems, e.g., metals and alloys, is also in doubt considering their distinctly different deformation mechanisms. Compared to polymers, it is far more challenging to construct bioinspired architectures in metallic materials. This is largely due to the difficulty in controlling the microstructures of metallic materials during traditional manufacturing processes, which usually involve high temperatures and high pressures. To date, most bioinspired metallic materials have been limited to those having nacre-like architectures which are fabricated by orienting non-isometric reinforcements, e.g., graphene and ceramic platelets, within metallic matrices by using powder-metallurgy techniques[11–17]. Exceptions have been realized recently, however, in the fabrication of fish-scale-like architectures in Cu-W and Mg-Ti systems by infiltrating a metal melt into fiber contextures of another metal with a higher melting point[18].

Additive manufacturing techniques of metals, as represented by three-dimensional (3-D) printing with selective laser or electron beam melting, offer a potent approach for processing metallic materials in a "bottom-up" fashion which resembles that in Nature. They are particularly effective in producing porous metallic scaffolds with complex pre-designed architectures, thereby bestowing new opportunities for implementing bioinspired designs. A variety of bioinspired architectures have been exploited in optimizing the mechanical properties of additive manufactured metallic scaffolds[19–23]. However, such architectural construction in dense metallic materials can be hampered by several limitations. First, the additive manufacturing techniques for metals have been largely limited to single material systems (or to a single ingredient even when it is composed of multiple components or constituents); the process can become markedly complicated when two or more types of ingredients are involved. In contrast, the majority of biological materials in Nature comprise at least two types of ingredients with distinctly different stiffness[24–26]. The relatively stiff and compliant ingredients are usually bicontinuous and topologically interconnected in 3-D space, forming specific interpenetrating-phase architectures[27,28]. Similar architectures have been constructed in polymeric composites by 3-D printing and proven to be effective in enhancing properties including stiffness, strength, fracture toughness, and energy dissipation ability, yet have rarely been realized in 3-D printed metallic materials[8–10]. Second, although a direct comparison of the mechanical properties among different types of bioinspired architectures is of vital importance for their selection in material design, this has yet to be achieved experimentally for metallic systems. Third, the structure-property relationships are the basis for the architectural optimization toward improved properties, but this is still largely unclear for bioinspired metallic materials with different architectures except for those mimicking nacre. In particular, the specific strengthening and toughening mechanisms associated with these architectures in metallic systems remain unexplored.

To address the above issues, we fabricated here a group of Mg-Ti interpenetrating-phase composites encompassing different bioinspired architectures using a two-step approach, specifically involving: (i) the 3-D printing of open porous Ti-6Al-4V scaffolds with bioinspired architectures, and (ii) the pressureless infiltration of the scaffolds with pure Mg melt. The Ti-6Al-4V alloy and Mg were selected as constituents owing to their high specific strengths associated with their low density and good biocompatibility[29–34]. In addition, they exhibit a large difference in stiffness which is qualitatively similar to that in biological materials. Three types of bioinspired architectures were designed: (i) the brick-and-mortar architecture mimicking nacre (e.g., of *Haliotis rufescens*[35,36]), (ii) the twisted plywood or the so-called Bouligand architecture mimicking arthropod exoskeleton (e.g., of *Odontodactylus scyllarus*[37,38]), and (iii) the crossed-lamellar architecture mimicking conch or bivalve shells (e.g., of *Saxidomus purpuratus*[39,40]). All these architectures, illustrated in Supplementary Fig. 1, have been shown to exhibit outstanding mechanical efficiency in their corresponding biological materials[35–43].

In this work, these three bioinspired architectures were constructed by 3-D printing Ti-6Al-4V scaffolds, which were then pressureless infiltrated with Mg to form Mg-Ti composites. This process utilizes the large difference in the melting points between the two constituents without severe chemical reaction, and the good wettability of the Mg melt with the Ti-6Al-4V alloy[44,45]. Our intent was to evaluate and compare the damage tolerance of these composites, specifically strength, toughness, and impact resistance (using the geometries shown in Supplementary Fig. 2), reveal their structure-property relationships, and clarify the toughening mechanisms associated with their specific bioinspired architectures. On this basis, we attempt to provide guidance for the architectural selection and design of bioinspired metallic materials. Further, we believe that our composites may also have the potential for structural and biomedical applications.

## Results

**Microstructural characteristics**. Figure 1 shows the 3-D X-ray diffraction topography (XRT) micrographs of the 3-D printed Ti-6Al-4V scaffolds and the infiltrated Mg-Ti composites with bioinspired brick-and-mortar (Fig. 1a), Bouligand (Fig. 1b), and crossed-lamellar architectures (Fig. 1c). The structures of their biological prototypes in Nature, respectively the shell of abalone *H. rufescens*[35,36], the dactyl club of mantis shrimp *O. scyllarus*[37,38], and the shell of bivalve mollusk *S. purpuratus*[39,40], are also presented for comparison. Despite the differences in their compositions and characteristic dimensions, the key features of the spatial arrangements in these biological materials were well reproduced into the scaffolds by 3-D printing and were retained in the composites after subsequent pressureless infiltration. Moreover, both constituents, Mg and Ti-6Al-4V, were continuous in the composites and were mutually interpenetrated in 3-D space. This also conforms to the common design motif in biological materials which generally demonstrate 3-D interpenetrating-phase architectures[27,28]. Specifically, the brick-and-mortar architecture was modified here by designing a hollow center within each Ti-6Al-4V unit in the scaffolds to aid the removal of extra metal powders after 3-D printing. As such, the Ti-6Al-4V units filled with Mg in the infiltrated composite can be regarded like the bricks in natural nacre. In detail, it is different from the real brick-and-mortar architecture of nacre, yet still reproduces its design motifs in the following aspects: (1) the hexagonal-shapes of structural units, (2) the staggered stacking of

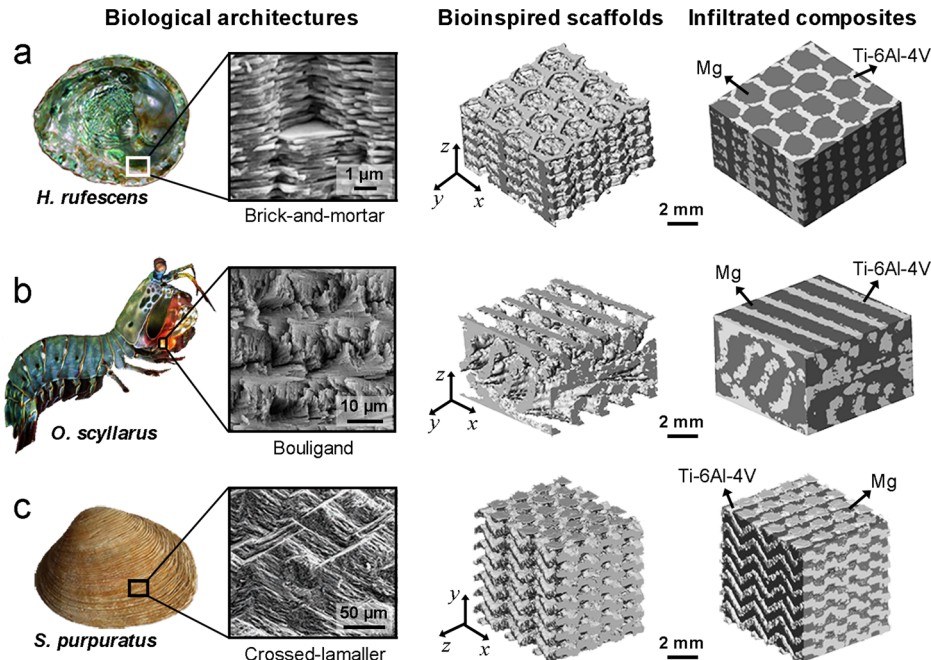

**Biological architectures**  **Bioinspired scaffolds**  **Infiltrated composites**

**Fig. 1 Bioinspired architectures of 3-D printed Ti-6Al-4V scaffolds and their infiltrated Mg-Ti composites.** The structures of representative biological prototypes in Nature, respectively the shell of abalone *H. rufescens*[36], the dactyl club of mantis shrimp *O. scyllarus*[38], and the shell of bivalve mollusk *S. purpuratus*[40] for the (**a**) brick-and-mortar, **b** Bouligand, and **c** crossed-lamellar architectures, are also presented for comparison. The architectures of the scaffolds are manifested using the 3-D XRT micrographs of the composites by filtering out the signals of Mg. The micrographs for the structures of biological materials are adapted with permission from refs. 36,38,40. Panel **a** adapted (with permission) from ref. 36. Panel **b** adapted (with permission) from ref. 38. Panel **c** adapted (with permission) from ref. 40.

units and the presence of interconnections between adjacent layers, and (3) the mutual interpenetration of stiff and soft phases in 3-D space.

The volume fractions of the Ti-6Al-4V phase for the bioinspired brick-and-mortar, Bouligand, and crossed-lamellar architectures were determined by XRT to be ~37.6 vol.%, ~34.7 vol.%, and ~52.8 vol.%, respectively. The densities of the composites can thus be approximated according to the rule-of-mixtures to be 2.78 g cm$^{-3}$, 2.70 g cm$^{-3}$, and 3.20 g cm$^{-3}$, respectively. These values were consistent with those measured experimentally, i.e., 2.78 g cm$^{-3}$, 2.70 g cm$^{-3}$, and 3.20 g cm$^{-3}$, indicating the virtual absence of pores in the composites. This was also verified by the fact that no pores were detected in XRT and scanning electron microscopy (SEM) imaging within the resolution ranges. In addition, residual stresses in the composites were low; measurements for the Ti-6Al-4V and Mg phases indicated values that were one to two orders of magnitude lower than their strengths (Supplementary Fig. 3), which may be associated with their slow cooling rate within the furnace after melt infiltration.

Figure 2 shows the fine microstructures, phase constitutions, and element distributions in Mg and Ti-6Al-4V constituents and at their interfaces, where the crossed-lamellar architecture is taken as an example (these characteristics were identical among the three architectures). The Ti-6Al-4V reinforcement was found to exhibit a fine basket-weave texture composed of α-Ti and β-Ti phases (Fig. 2a, b). Such a structure has been shown to be effective for arresting crack propagation in the alloy, thereby favoring improved fracture toughness[46]. Indeed, the Ti-6Al-4V phase exhibits similar microstructures and mechanical properties (as determined by nanoindentation testing) before and after melt infiltration, i.e., in the 3-D printed scaffolds and infiltrated composites (Supplementary Fig. 4). This indicates that the material properties of Ti-6Al-4V phase were essentially unaffected by the infiltration process.

After infiltration, new phases were formed in the Mg matrix mainly at the grain boundaries and at the interfaces with Ti-6Al-4V reinforcement (Fig. 2a, c). These phases were determined by microzone X-ray diffraction (XRD) to be $TiSi_2$ and $Ti_5Si_3$ (Fig. 2d), which were the products of the reactions between the Si element contained in Mg (which was produced by silicothermic reduction method where silicon was used as a reducing agent[47]) and the Ti element dissolved from the Ti-6Al-4V constituent. A small amount of AlMg phase emerged in the grains of the Mg matrix as a result of the diffusion of elemental Al from Ti-6Al-4V to the Mg and its reaction with Mg. Energy-dispersive X-ray spectroscopy (EDS) measurements indicated a nearly uniform distribution of Al in the Mg matrix (Fig. 2e). By comparison, the Ti, Si, and V elements were relatively enriched, along with the depletion of elemental Mg, at the grain boundaries of the Mg matrix and the interfaces between Mg and Ti-6Al-4V constituents. In addition, the Ti-6Al-4V and Mg phases demonstrated a microscopically rough interface with the mixing of material over a range of ~100 μm, as evidenced by the gradual changes in the contents of Mg and Ti elements (Supplementary Fig. 5). Such features have been shown to be beneficial for enhancing the interfacial bonding between constituents[48]. The strong interfacial bonding here can be verified by the fact that fracture occurred in the Mg phase rather than along the interface for the combination of Ti-6Al-4V and Mg phases subject to tensile loading (Supplementary Fig. 6).

**Tensile properties**. Figure 3a shows the representative quasi-static tensile stress-strain curves of the bioinspired composites with their loading configurations illustrated in the insets. The data for the pure Mg solidified from the infiltration temperature along with the composites, the 3-D printed dense Ti-6Al-4V alloy and the porous Ti-6Al-4V scaffolds with different architectures are also presented for comparison in Supplementary Fig. 7. The

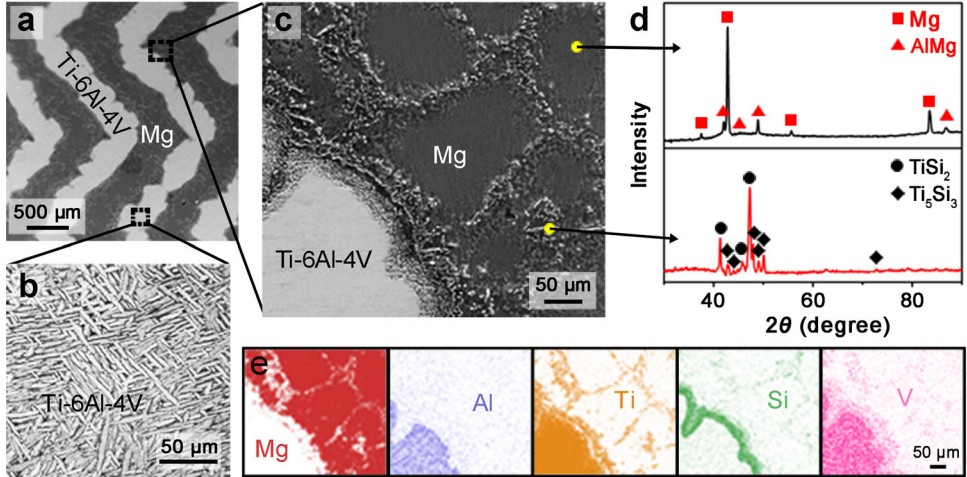

**Fig. 2 Fine microstructures, phase constitutions, and element distributions in bioinspired Mg-Ti composites. a–c** Micrographs of **a** the Mg-Ti composite, **b** the Ti-6Al-4V reinforcement, and **c** the Mg matrix along with the interfacial region between Mg and Ti-6Al-4V phases. **d** Microzone XRD patterns of the Mg matrix within the grains (upper) and at the grain boundaries (lower). **e** Area distributions of the elemental Mg, Al, Ti, Si, and V obtained by EDS measurement in the region corresponding to the image in **c**.

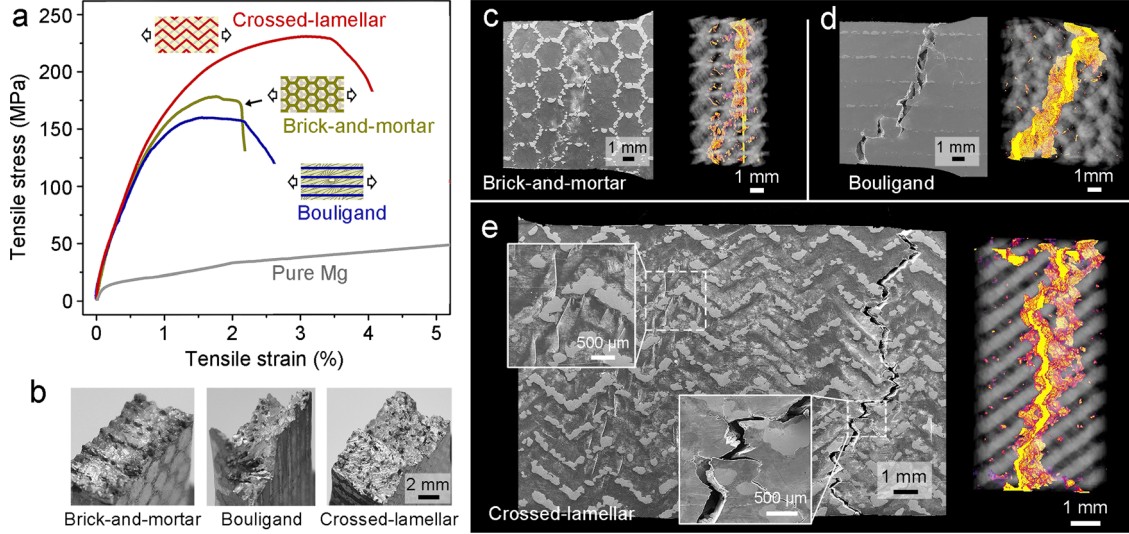

**Fig. 3 Uniaxial tensile properties and damage characteristics of the bioinspired Mg-Ti composites with three different architectures. a** Tensile engineering stress-strain curves and **b** overall fracture morphologies of the composites with the loading configurations for tensile tests illustrated in the insets in (**a**). The data for coarse-grained pure Mg are also shown in (**a**) for comparison. **c–e** SEM images and X-ray computed tomography (CT) volume renderings of the tensile samples unloaded just prior to fracture for the (**c**) brick-and-mortar, (**d**) Bouligand, and (**e**) crossed-lamellar architectures. The CT images were processed by filtering out the signals of Mg and highlighting the cracking regions in yellow color. The insets in **e** magnify the slip bands within Mg and the deflection and branching of cracks in the composite with crossed-lamellar architecture.

tensile properties, including the Young's modulus $E$, yield strength at 0.2% offset strain $\sigma_{YS}$, ultimate tensile strength $\sigma_{UTS}$, elongation to fracture $\varepsilon_F$, and the work of fracture $U_F$ (determined from the area under stress-strain curve until fracture) are listed in Table 1. To better embody the effects of architecture, some of these properties are normalized with respect to the density by taking into account their differences in phase constitution. The bioinspired composites exhibit markedly higher strengths than pure Mg and the Ti-6Al-4V scaffolds along with a decrease in ductility as compared to pure Mg. The strength and ductility are similar for the brick-and-mortar and Bouligand architectures, although the crossed-lamellar architecture demonstrates a synergistic elevation in properties, leading to an increase in the work of fracture by ~2.7 times that of the two other architectures. Such property enhancements in the latter structure are still

apparent even when they are normalized by the density, clearly implying the notable mechanical efficiency of the crossed-lamellar architecture. Nevertheless, changes in the Young's modulus are insignificant among the three types of architectures.

Although the tensile properties of the bioinspired composites are inferior to those of the 3-D printed dense Ti-6Al-4V alloy, they can be compared with a compositionally similar Mg alloy composite reinforced with Ti-6Al-4V particles that was fabricated by means of stir casting[49]. This composite possessed a comparable phase constitution (with ~50 vol.% Mg) as the bioinspired materials described here, but was not developed with any specific architectures; instead, the Ti-6Al-4V particles were uniformly dispersed within the Mg matrix. Fracture readily occurred in this composite under tensile loading at a low-stress level of 116–160 MPa with a limited total elongation of less than

**Table 1 Mechanical properties of bioinspired Mg-Ti composites with different architectures.**

| Mechanical properties | Bioinspired Mg-Ti composites | | | Ti-6Al-4V scaffolds | | | Pure Mg | Dense Ti-6Al-4V alloy | Uniform composite[49] |
|---|---|---|---|---|---|---|---|---|---|
| | Brick-and-mortar | Bouligand | Crossed-lamellar | Brick-and-mortar | Bouligand | Crossed-lamellar | | | |
| $E$ (GPa) | 58.2 ± 4.6 | 54.0 ± 5.1 | 55.5 ± 6.3 | 2.3 ± 0.3 | 1.5 ± 0.2 | 2.6 ± 0.1 | 40.2 ± 1.0 | 109.8 ± 2.8 | |
| $\sigma_{YS}$ (MPa) | 77.2 ± 6.5 | 69.2 ± 2.4 | 110.0 ± 3.0 | 18.2 ± 1.0 | 9.6 ± 0.3 | 18.5 ± 0.4 | 16.5 ± 1.6 | 923.5 ± 2.6 | |
| $\sigma_{UTS}$ (MPa) | 169.7 ± 7.9 | 156.6 ± 8.6 | 226.6 ± 5.0 | 36.7 ± 1.2 | 12.2 ± 0.1 | 65.4 ± 0.4 | 55.9 ± 5.2 | 994.4 ± 11.3 | 116–160 |
| $\varepsilon_F$ (%) | 2.09 ± 0.05 | 2.14 ± 0.13 | 3.95 ± 0.04 | 3.16 ± 0.18 | 1.00 ± 0.04 | 6.27 ± 0.17 | 8.82 ± 0.34 | 16.19 ± 1.68 | ≤0.5 |
| $U_F$ (J cm$^{-3}$) | 2.61 ± 0.20 | 2.60 ± 0.24 | 7.09±0.27 | 0.79 ± 0.07 | 0.081 ± 0.004 | 2.92 ± 0.46 | 3.47 ± 0.32 | 153.29 ± 16.93 | 0.58–0.80 |
| $\sigma_Y/\rho$ (MPa (g cm$^{-3}$)$^{-1}$) | 27.8 ± 2.3 | 25.6 ± 0.9 | 34.4 ± 1.0 | 10.7 ± 0.6 | 6.1 ± 0.2 | 7.8 ± 0.2 | 9.5 ± 0.9 | 204.8±0.6 | |
| $\sigma_{UTS}/\rho$ (MPa (g cm$^{-3}$)$^{-1}$) | 61.0 ± 2.9 | 58.0 ± 3.2 | 70.8 ± 1.6 | 21.7 ± 0.7 | 7.8 ± 0.1 | 27.5 ± 0.2 | 32.1 ± 3.0 | 220.5 ± 2.5 | 37–51 |
| $U_F/\rho$ (J g$^{-1}$) | 0.94 ± 0.07 | 0.96 ± 0.09 | 2.22 ± 0.08 | 0.47 ± 0.04 | 0.052 ± 0.003 | 1.23 ± 0.19 | 2.10 ± 0.19 | 33.99 ± 3.75 | 0.19–0.26 |
| $K_{Ic}$ (MPa m$^{0.5}$) | 22.4 | 25.0 | 27.7 | | | | 6–20 | 79.6 | |
| $J_{Ic}$ (kJ m$^{-2}$) | 8.9 | 11.3 | 13.0 | | | | | 57.9 | |
| $J_{ss}$ (kJ m$^{-2}$) | 15.4 | 22.7 | 30.2 | | | | | 82.4 | |
| $K_{ss}$ (MPa m$^{0.5}$) | 31.8 | 37.0 | 43.5 | | | | | 101.2 | |

The data for the Ti-6Al-4V scaffolds without infiltration of Mg, the coarse-grained pure Mg, the 3-D printed dense Ti-6Al-4V alloy, and the Mg alloy composite reinforced with discrete Ti-6Al-4V particles (uniform composite) are also presented for comparison. $E$: Young's modulus; $\sigma_{YS}$: yield strength; $\sigma_{UTS}$: ultimate tensile strength; $\varepsilon_F$: elongation to fracture; $U_F$: work of fracture; $\rho$: density; $K_{Ic}$: K-based fracture toughness; $J_{Ic}$: J-based fracture toughness; $K_{ss}$: Crack-growth toughness at $\Delta a = 1mm$; $K_{ss}$: K-based crack-growth toughness at $\Delta a = 1mm$.

0.5%; the specific ultimate tensile strength ($\sigma_{UTS}$) was found to be 37–51 MPa (g cm$^{-3}$)$^{-1}$. The much superior mechanical properties of the current composites indicate an effective role of the bioinspired architectures, in particular the crossed-lamellar architecture, in enhancing the strength and ductility.

The bioinspired composites show large differences in fracture morphologies depending on their specific architectures, as shown in Fig. 3b. Specifically, the fracture surface was relatively flat for the brick-and-mortar architecture but showed decorations with rows of broken edges of the Ti-6Al-4V bricks. The fracture surfaces clearly exhibited increased tortuosity for the Bouligand and crossed-lamellar architectures, respectively along a helical trajectory and periodically zig-zag paths for the former and latter —both are consistent with the spatial arrangement of their constituents. The deformation and damage characteristics can be clearly appreciated in Fig. 3c–e by the 2-D and 3-D micrographs of samples that were unloaded just prior to fracture. The brick-and-mortar architecture exhibited a large degree of localization of plastic deformation and cracks in relatively narrow regions in the plane that was nearly normal to the tensile direction (Fig. 3c). For the Bouligand architecture, the cracking paths were helicoidally twisted among layers with different fiber orientations (Fig. 3d); nevertheless, within an individual layer, the plastic deformation was still concentrated near the cracks.

In comparison, the crossed-lamellar architecture exhibited the broadest (non-localized) distribution of plastic deformation and damage extending over the entire gauge section (Fig. 3e). Abundant slip bands were formed within the Mg matrix, but were localized between adjacent Ti-6Al-4V platelets (inset). Moreover, the cracking paths were tortuous, both within the constituent layer (in-plane) and across different layers (out-of-plane), and were complicated by a number of branches at their deflecting points (inset). All these features suggest a more effective role of the crossed-lamellar architecture in delaying the final fracture of the composite, conforming well to its best-displayed combination of strength and ductility. By contrast, the plastic deformation was largely localized within the Mg matrix in the Mg alloy composites reinforced by discrete Ti-6Al-4V particles; cracking readily tended to occur and develop along the interfaces between the two phases[29–34,49]. As such, the enhanced mechanical properties of the bioinspired composites are closely associated with the delocalization of damage and the arrest of crack extension by the 3-D interpenetration of constituents and their specific architectures.

**Laminate theory analysis.** In the following section, classical laminate theory[50] is adapted for the bioinspired composites by incorporating their architectural characteristics in order to establish quantitative structure-property relationships for their Young's modulus and strengths. As shown in Fig. 4a, for a laminate composite subject to uniaxial in-plane tension, the strains are identical among the laminae which represent the constituent layers of the composite. Therefore, the Young's modulus of the entire laminate can be obtained from those of the individual laminae according to the rule-of-mixtures based on the Voigt model[50,51]. The yield and ultimate tensile strengths can also be accessed using the specific stresses corresponding to the failure of, respectively, the first and the final laminae, i.e., by assuming that the yielding and fracture of the laminate are caused by the failure of these laminae[18,50,52,53]. Specifically, these properties of the laminate can be approximated using those of the individual lamina for the brick-and-mortar and crossed-lamellar architectures, because the inclinations of constituents with respect to the loading direction are essentially equivalent in all the laminae despite the staggering between them. In this context, it is important to formulate the mechanical properties of each individual lamina.

The structure of the lamina in the bioinspired composites is determined essentially by the orientations of their constituents. Such characteristics can be described by extracting an elementary unit where the constituents are unidirectionally aligned, as illustrated in Fig. 4a by taking the Bouligand architecture as an example. The local coordinate system for the element is denoted by (1, 2) with the principal axes 1 and 2 defined to be parallel and perpendicular to the alignment of its constituents (Supplementary Fig. 8). The global coordinate system for the entire lamina is denoted by (L1, L2) where the principal axes L1 and L2 are, respectively, parallel and perpendicular to the tensile direction. Both coordinate systems are right-handed and exhibit an inclination angle $\theta$ between them. The laminae are assumed to be perfectly bonded in the laminate such that the slipping deformation between them can be ignored. This is consistent with the experimental result that the layers were indeed welded

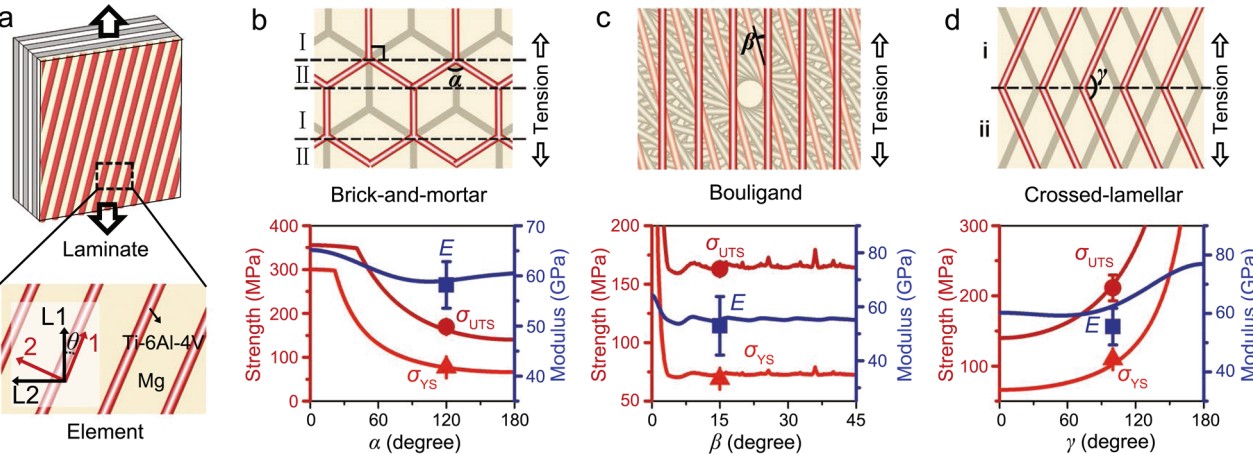

**Fig. 4 Laminate theory analysis for the Young's modulus and strengths for the three types of bioinspired architectures. a** Schematic illustration of the elementary structural unit extracted from an individual lamina in the laminate subject to in-plane uniaxial tension. The local coordinate system for the element and the global coordinate system for the lamina are denoted, respectively, using (1, 2) and (L1, L2). **b–d** Schematic illustrations of the spatial arrangements of constituents for the bioinspired (**b**) brick-and-mortar, (**c**) Bouligand, and (**d**) crossed-lamellar architectures, and the theoretical results on the variations in the Young's modulus $E$, yield strength $\sigma_{YS}$ and ultimate tensile strength $\sigma_{UTS}$ of the composites as a function of the specific angles $\alpha$, $\beta$ and $\gamma$. The experimental data (solid dots) are also presented for comparison.

together by the fusion points formed during 3-D printing; indeed, no obvious slippage was detected between layers in the composites.

For the Bouligand architecture, the orientations of constituents are consistent within each individual lamina, conforming well to the above structural element (Fig. 4a, c). The twist angle between the reinforcement fibers in adjacent layers is denoted by $\beta$. For the brick-and-mortar and crossed-lamellar architectures, however, the constituents show varying orientations across the lamina and as a result cannot be described using a single element. Instead, the lamina can be regarded as a combination of two types of structural units which are alternately arranged and serially connected along the loading direction, as shown in Fig. 4b, d. As such, the Young's modulus of the lamina can be accessed from those of the units according to the Reuss model considering that the stresses are identical between them[51]. The strengths of the lamina can also be approximated using the lower ones of the units[54]. Specifically for the brick-and-mortar architecture, the edges of bricks are parallel to the tensile direction in element I, but are inclined in a symmetric fashion by a specific angle $\alpha/2$ ($\alpha$ denotes the apex angle of the hexagonal bricks) in element II (Fig. 4b). Similarly, the lamina can be divided into elements i and ii where the platelets show opposite orientations with respect to the tensile direction for the crossed-lamellar architecture (Fig. 4d). As such, this architecture can be described using the inclination angle between platelets in the two elements $\gamma$. In this scenario, the mechanical properties (Young's modulus and strengths) of the entire laminate can be formulated following the scheme in sequence as "element-lamina-laminate".

Because the characteristic dimensions of the constituents are similar, the mechanical properties of the composites with different architectures stem essentially from their spatial arrangements which can be determined by the specific angles $\alpha$, $\beta$, and $\gamma$. Specifically, for the element II of the brick-and-mortar architecture, the neighboring edges of bricks are systematically inclined (by $\pm\alpha/2$) with respect to the tensile direction, and are thus equivalent in terms of their effects on the Young's modulus and strengths of the lamina. Under plane-strain conditions, for an individual element where the reinforcements are unidirectionally aligned, the Young's moduli along the axes 1 and 2 can be accessed from those of the constituents according to the rule-of-mixtures in line with the Voigt and Reuss models[50,51],

respectively. Accordingly, the Young's modulus of the element along the tensile direction can be obtained by transforming the stiffness matrices of the local coordinate system (1, 2) into the global one (L1, L2)[18,50,53]. In addition, the strengths (yield strength and ultimate tensile strength) of an individual element along the tensile direction can be accessed using those along the principal axes 1 and 2 following the Tsai-Hill failure criterion[55–57]. This criterion has been proven to be applicable to a wide range of composites, including those with bioinspired architectures[18,50,51,53–56]. The strengths along the 1 axis can be seen as the weighted averages of those constituents by their volume fractions, similar to the case for the Young's modulus. The strengths along the 2 axis and the shear strength along the 1-2 direction can be approximated using those of coarse-grained pure Mg considering that the failure of composite in these loading configurations is principally governed by the relatively weak matrix. By this means, the Young's modulus and strengths of the individual lamina can be formulated from those of the elements; this further allows for the determination of the properties for the entire laminate. Full details about the derivation can be found in Supplementary Notes 1–3 in the Supplementary Materials.

The analytical results for the variation in Young's modulus and strength as a function of the specific angles for the three types of bioinspired architectures (with the same phase constitutions as the experiments) are shown in Fig. 4b–d. The experimental data can be well described by the theoretical modeling, clearly indicating a good validity of the above analysis and the derived structure-property relationships. We note here that the yield strength and ultimate tensile strength of the structural element along the 2 axis were set by data fitting to be 70 MPa and 140 MPa, respectively, for all the composites. These values are higher than those of the as-cast pure Mg with coarse grains[57]. This is caused by the presence of Ti-6Al-4V interconnections throughout the Mg matrix and the precipitation of intermetallic phases within it, particularly at its grain boundaries, in the composites.

Specifically, for the brick-and-mortar architecture (Fig. 4b), the strength of the composite remains relatively stable as the apex angle of the hexagonal bricks $\alpha$ increases, before displaying a decreasing trend when $\alpha$ exceeds ~25° and ~40°, respectively, for the yield strength and ultimate tensile strength. Such a transition is caused by the fact that the yielding and fracture of the

composite are prone to occur in element I at a lower $\alpha$, but in element II when $\alpha$ exceeds these specific values (Supplementary Note 1). By contrast, the Young's modulus initially decreases and then exhibits a gradual increasing trend with the increase in $\alpha$. For the Bouligand architecture (Fig. 4c), the Young's modulus and strength decrease rapidly with the initial increase in the twist angle $\beta$, and then show fluctuations over the entire $\beta$ range with the appearance of some peaks for the strengths at specific $\beta$ values. This is caused by the variations in the overall deviation of fibers with respect to the loading direction and the overlapping of fibers between different laminae at these $\beta$ values, as shown in Supplementary Fig. 9. For the crossed-lamellar architecture (Fig. 4d), the Young's modulus shows a gradual decreasing trend until reaching its minimum at a specific $\gamma$ of ~58° and then increases with the further increase in $\gamma$. The strengths exhibit a monotonic rising trend with $\gamma$ over its entire range. With respect to the effects of phase constitution, the Young's modulus increases monotonically with increasing Ti-6Al-4V content for all the three types of architectures when the specific angles are fixed (Supplementary Fig. 10). In addition, despite the differences in absolute values, the Young's modulus and strengths demonstrate similar varying trends with the specific angles for these architectures when the phase constitutions are fixed (Supplementary Fig. 11). Although deviations may arise as a result of the possible interfacial shearing between constituents or their reorientation during deformation[18,58], the above theoretical results offer some guidance for tailoring the mechanical properties of the bioinspired composites by manipulating their architectures.

**Fracture and impact toughnesses.** Figure 5a shows the variations in $J$-integral as a function of the crack extension $\Delta a$ for the bioinspired composites with different architectures with the specific loading configurations illustrated in the insets. $J$-based crack-resistance curves (R-curves) were obtained by fitting the valid $J$-$\Delta a$ data (solid dots) using the power-law relationship $J = C_1(\Delta a)^{C_2}$ where $C_1$ and $C_2$ are fitting parameters, in accordance with the standard ASTM E1820[59]. It is seen that the R-curves rise constantly as the crack extends for all the composites, but show a decrease in the rising rate as manifested by the slope of the curve, demonstrating a varying trend in a "Γ"-shaped form[7,59]. This is similar with the case reported for the 3-D printed polymeric

composites with bioinspired brick-and-mortar and crossed-lamellar architectures[7]. However, the polymeric composite with a rotating plywood architecture (i.e., Bouligand architecture) displayed "J"-shaped R-curves where the rising rate of $J$-integral increases with crack extension[7]. Such behavior can be attributed to the fact that the polymeric composite exhibited relatively low resistance to the onset of crack extension (much lower than metals), but became increasingly more resistant to crack growth owing to the notable crack deflection and twisting along its architecture; the crack tip clearly deviated from a mode I stress state. For the current bioinspired metal composites, both the $J$-based R-curves and their variation are closely dependent on the specific architectures. The crossed-lamellar architecture leads to the highest fracture resistance along with the largest rising R-curve over the entire range of crack extension. This is followed by the Bouligand architecture, whereas the $J$-based R-curve of the brick-and-mortar architecture is the lowest and rises the most slowly.

The provisional fracture toughness $J_Q$ was determined from the intersection of the R-curves and the 0.2-mm offset lines (dashed lines) with a slope of $(\sigma_{YS} + \sigma_{UTS})$ in accordance with ASTM E1820[59]. The $J_Q$ was validated to be the size-independent plane-strain fracture toughness $J_{Ic}$ for all the composites as the validity requirements of $(W - a_0), B > 20 J_Q/(\sigma_{YS} + \sigma_{UTS})$ were met. The stress intensity $K$-based crack-initiation fracture toughness, $K_{Ic}$, was determined directly from the load-displacement curves in accordance with ASTM E399[60]. The crack-growth fracture toughness $J_{SS}$ was represented using the $J$-integrals at crack extension of 1 mm[59]. The corresponding $K$-based fracture toughness, $K_{SS}$, can be accessed from $J_{SS}$ according to the mode I $J$-$K$ equivalence relationship of $K = (EJ/(1 - \nu^2))^{1/2}$ where $E$ and $\nu$ are the Young's modulus and Poisson's ratio. All these fracture toughness values are listed in Table 1. The fracture toughnesses of these composites can be seen to lie between those of pure Mg and 3-D printed dense Ti-6Al-4V alloy (Supplementary Fig. 12), but are markedly higher than those reported for the 3-D printed polymeric composites with bioinspired architectures[6,7,10]. In addition, it is clear that the crossed-lamellar architecture is consistently the toughest among the three types of bioinspired architectures.

Figure 5b presents the impact toughnesses of the bioinspired composites loaded along different directions, as illustrated in the insets. Statistically significant differences in impact toughness can only be found for the crossed-lamellar architecture among

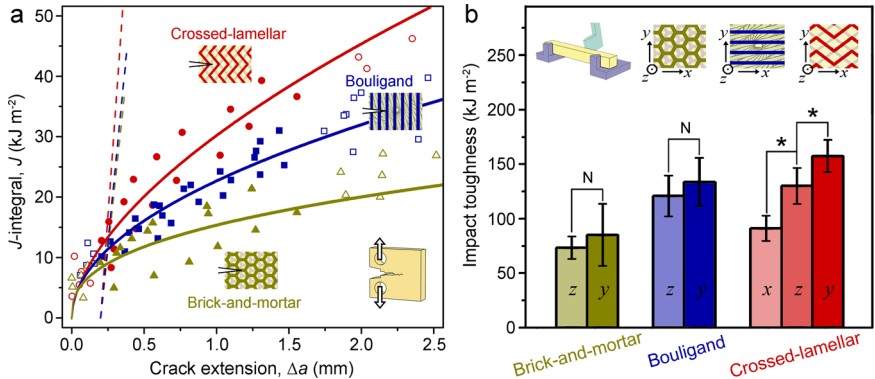

**Fig. 5 Fracture and impact toughnesses of the bioinspired Mg-Ti composites with different architectures. a** Variations in $J$-integral as a function of the crack extension $\Delta a$ along with the fitted R-curves for the composites. The loading configurations and cracking directions with respect to the architectures are illustrated in the insets. The dash lines for determining the fracture toughnesses have a slope of $(\sigma_{YS} + \sigma_{UTS})$ and correspond to a 0.2-mm offset strain. **b** Impact toughnesses of the bioinspired composites when loaded along different directions as illustrated in the insets. The specific loading directions, which are clearly illustrated in Supplementary Fig. 2, are marked in the columns. Error bars represent standard deviation. The symbols "★" and "N" indicate that the differences are statistically significant and insignificant, respectively, with a significance level of $P$ value < 0.05 tested using two-tailed Student's $t$ test.

different directions, indicating distinct anisotropy. Specifically, the highest impact toughness is generated when the impact load is parallel to the boundaries between lamellae (i.e., along the $y$ axis in $y$-$z$ plane). This is exactly the most frequent (in-service) loading configuration in its natural prototype of the bivalve shell of *S. purpuratus*, e.g., caused by attack from a mantis shrimp[37,38]. The impact toughnesses do not show statistically significant differences among different directions for either the brick-and-mortar or the Bouligand architectures. Nevertheless, in general the crossed-lamellar and Bouligand architectures are tougher than the brick-and-mortar one under impact loading conditions. The only exception exists for the crossed-lamellar architecture when the impact load is perpendicular to the boundaries of lamellae (i.e., along the $x$ axis in $x$-$z$ plane).

Figure 6 shows the cracking morphologies of the quasi-static fracture toughness samples for the three types of bioinspired composites. The trajectories of cracks and their interactions with architectures in 3-D space can be visualized by filtering out the signals of the constituents in the X-ray computed tomography (CT) images, also shown in Supplementary Movies 1–3. Crack propagation in all the composites clearly deviated from a straight path, occurring in both in-plane (crack deflection) and out-of-plane (crack twist) modes. In addition, all the cracks were bridged by a number of uncracked ligaments behind their tips, as manifested by the dark domains involved in the cracking profiles. However, the extents of crack deflection/twist and bridging were markedly different among these composites depending on their specific architectures. The cracks tended to penetrate the brick walls for the brick-and-mortar architecture, resulting in limited deflection and twist (Fig. 6a). Also, the crack bridging only occurred in a narrow region near the crack tip. The Bouligand architecture exhibited more obvious crack twist because of the helicoidal alignment of Ti-6Al-4V fibers among layers (Fig. 6b). Crack bridging was also promoted owing to microcracking ahead of the crack tip, as indicated by the arrows, when the fibers were inclined from the cracking plane. In comparison, the crossed-lamellar architecture displayed the broadest cracking area in 3-D space owing to its most notable evidence of crack deflection and twist (Fig. 6c). In addition to the changes in the cracking paths across lamellae or layers, cracks propagated along a zig-zag trajectory even within the same lamella as they penetrated

different platelets (inset). A large number of uncracked ligaments were formed in the cracking profile; they acted to carry load which otherwise would be used to promote crack extension. The crack opening was also limited by the frictional sliding between crack faces and the pullout or rupture of the ligaments. All these features indicate the most notable toughening efficiency of the crossed-lamellar architecture among the three bioinspired ones, conforming well to its highest measured fracture toughness.

## Discussion

Figure 7 shows a direct comparison of mechanical properties among the three types of bioinspired composites with different architectures. The data for Mg alloy composite reinforced with discrete Ti-6Al-4V particles are also presented for comparison[49]. The yield strength ($\sigma_{YS}$), ultimate tensile strength ($\sigma_{UTS}$), and work of fracture ($U_F$) are normalized by density ($\rho$) to compensate for the effects of their differences in phase constitution. The bioinspired architectures clearly endow the composites with higher strengths and work of fracture (as compared to the composites without bioinspired architectures). In addition, they play an effective role in delocalizing damage and resisting crack propagation under both quasi-static and dynamic loading conditions, bestowing improved ductility along with notable fracture and impact toughness properties.

The good mechanical properties of the bioinspired composites stem primarily from the following two aspects: First, the bicontinuous nature and interpenetration of the Mg and Ti-6Al-4V constituents in 3-D space allow for an effective stress transfer within and between each phase, thereby conferring a high strengthening efficiency of the reinforcement[61–63]. The deformation and cracking of the relatively weak Mg phase can also be restricted by the mutual partition of constituents; as a result the fracture of the entire composite can be retarded. Second, the specific bioinspired spatial arrangements of constituents create a series of extrinsic toughening mechanisms in the composites, specifically crack deflection/twist and uncracked-ligament bridging. These mechanisms act principally behind the crack tip and function to shield the crack tip from the applied stress[64]. Microcracks are also prone to emerge ahead of the crack tip as the crack inevitably encounters the Ti-6Al-4V reinforcement. This can direct the growing crack to deflect and twist and

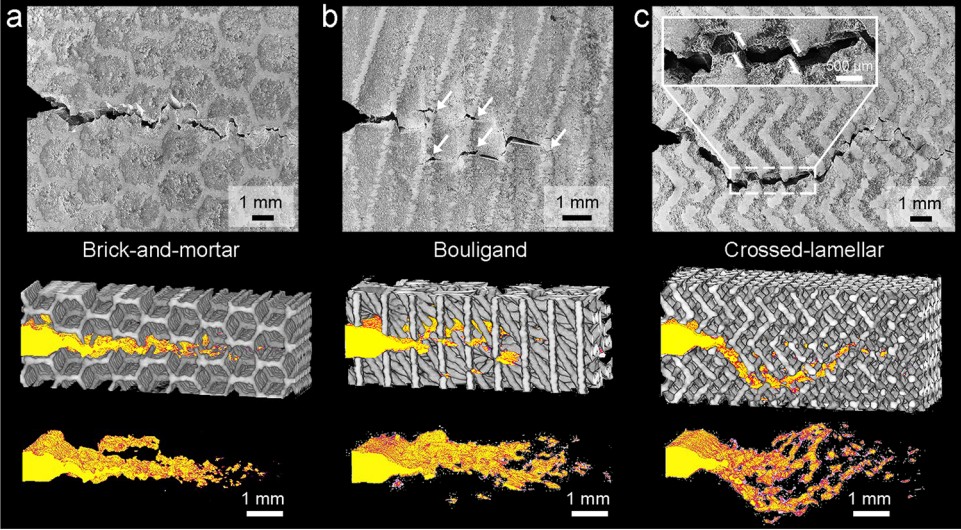

**Fig. 6 Cracking morphologies and toughening mechanisms in the bioinspired Mg-Ti composites.** SEM images and CT volume renderings of the quasi-static fracture toughness samples for the composites with bioinspired (**a**) brick-and-mortar, (**b**) Bouligand, and (**c**) crossed-lamellar architectures. The CT images were processed by filtering out the signals of constituents and highlighting the cracking regions. The white arrows in (**b**) indicate the microcracks ahead of the crack tip. The inset in (**c**) magnifies the zig-zag cracking path and the resulted frictional sliding between crack faces as indicated by the arrows.

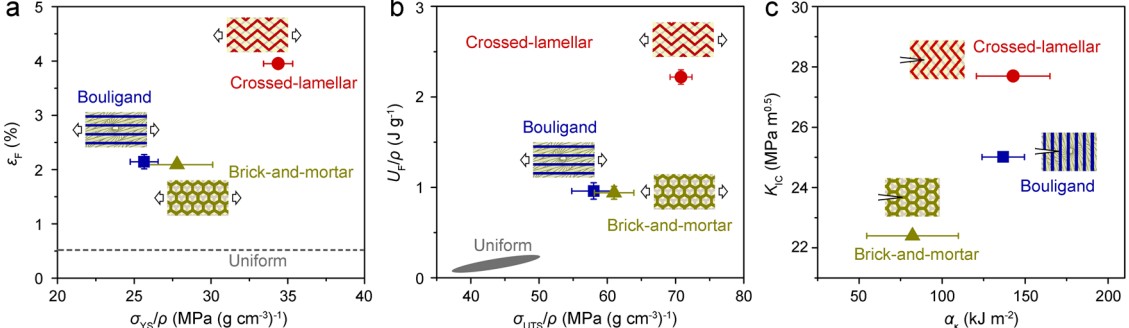

**Fig. 7 Comparison of mechanical properties for the bioinspired Mg-Ti composites with different architectures. a** Elongation to fracture $\varepsilon_F$ versus specific yield strength $\sigma_{YS}/\rho$; **b** specific work of fracture $U_F/\rho$ versus specific ultimate tensile strength $\sigma_{UTS}/\rho$; **c** fracture toughness $K_{Ic}$ versus impact toughness $\alpha_\kappa$. The loading configurations with respect to the architectures for uniaxial tension, fracture toughness and impact toughness measurements are illustrated in the insets. The data for the Mg alloy composite reinforced with discrete Ti-6Al-4V particles (uniform) are also shown for comparison[49].

produce uncracked ligaments behind the crack tip, and by this means promote the above extrinsic toughening mechanisms. In addition, the interconnection between constituent layers, realized through the fusion joints formed during 3-D printing, is critical for avoiding the delamination of the composites, thereby ensuring the above strengthening and toughening effects of the architectures. In contrast, cracking is mainly localized within the Mg matrix and at the interfaces between constituents in the composites without bioinspired architectures[29–34,49]. Contributions to the fracture toughness are thus mainly developed from intrinsic toughening mechanisms which are related to the plasticity of the constituents[64].

With regard to the bioinspired composites, the mechanical properties demonstrate a strong dependence on the types and detailed characteristics of their architectures. In particular, the crossed-lamellar architecture exhibits simultaneously the highest levels of strength, elongation to fracture, work of fracture, as well as fracture and impact toughnesses. This can be principally attributed to its hierarchical nature which does not exist in the present brick-and-mortar or Bouligand architectures[35–38]. The orientations of constituents are constantly varied at three levels of hierarchy, i.e., corresponding respectively to the platelet, lamella, and layer. Abundant interfaces are also present at all these length scales. In comparison, the (specific) strengths, elongation to fracture, and work of fracture are all comparable (without statistically significant differences) between the brick-and-mortar or Bouligand architectures. Nevertheless, the Bouligand architecture exhibits a markedly higher toughening efficiency than the brick-and-mortar one owing to the enhanced role of crack twist and uncracked-ligament bridging. In addition, the Young's modulus and strengths of all these bioinspired architectures can be quantitatively described based on the structure-property relationships established here. This was realized by adapting the classical laminate theory to incorporate their architectural characteristics and on the description of these architectures using the key specific angles $\alpha$, $\beta$, and $\gamma$, as described in Supplementary Notes 2–4.

The notable mechanical properties along with the low density of the bioinspired Mg-Ti composites endow them with the potential for structural application; the crossed-lamellar architecture is particularly promising in this respect. For example, these materials may serve as new biomedical materials with good mechanical behavior, in part due to the good biocompatibility of their constituents[33,34]. It is noted that the properties of these bioinspired composites can definitely be further optimized, e.g., by regulating the architectural parameters and by tailoring the chemical characteristics of constituents. In addition, the current processing route, i.e., by 3-D printing and melt infiltration, demonstrates a good feasibility for fabricating bioinspired

composites in metallic systems. In particular, the 3-D printing techniques enable a precise construction of bioinspired architectures exactly in line with pre-designed models. The established structure-property relationships may provide the theoretical basis for the design of these models, especially for selecting architectural types and determining their detailed orientation characteristics. In this scenario, it may become possible to exploit new bioinspired composites more efficiently following the iterative scheme as "theoretical analysis—architectural design—composite fabrication" to better fulfill the property requirements of materials.

In conclusion, three types of Mg-Ti composites with bioinspired brick-and-mortar, Bouligand and crossed-lamellar architectures were fabricated by pressureless infiltration of pure Mg melt into 3-D printed Ti-6Al-4V scaffolds. By characterizing and comparing their microstructural characteristics, uniaxial tensile properties, and fracture and impact toughnesses, analyzing their structure-property relationships based on laminate theory modeling, and exploring their toughening mechanisms associated with specific architectures, the following conclusions can be drawn:

1. The Mg and Ti-6Al-4V constituents were both continuous, mutually interpenetrated in 3-D space, and exhibited specific spatial arrangements in the bioinspired composites qualitatively consistent with the architectures in their natural prototypes. The adjacent layers of reinforcement were interconnected by the fusion joints formed during 3-D printing. Precipitates emerged at the grain boundaries of Mg and the interfaces between constituents.
2. The tensile properties of the bioinspired composites all outperform those of composites reinforced by discrete Ti-6Al-4V particles, yet are closely dependent on their specific architectures. The Young's modulus and strengths of the composites can be formulated by modifying the classical laminate theory to incorporate their architectural characteristics, especially associated with the orientations of their constituents. This may offer a theoretical basis for the architectural selection and design of bioinspired composites.
3. The bioinspired composites exhibited "Γ"-shaped R-curves which display a rising trend of J-integral but a decrease in rate as the crack extends. The bioinspired architectures induced a series of extrinsic toughening mechanisms, including crack deflection/twist and uncracked-ligament bridging, to shield the crack tip from the applied stress. These mechanisms were additionally promoted by micro-cracking ahead of the crack tip. Only the crossed-lamellar architecture exhibited significant anisotropy in impact toughness when loaded along different directions.

4. The crossed-lamellar architecture was the most effective among the three bioinspired architectures in strengthening materials, delocalizing damage and resisting crack extension. This architecture endowed the composites with the best combination of mechanical properties, including strength, elongation to fracture, work of fracture, and fracture and impact toughnesses. This was largely attributed to its hierarchical nature where the variations in constituent orientations and interfaces are active at different length scales.

The bioinspired Mg-Ti composites may have the potential for structural and biomedical applications. The notable strengthening and toughening efficiencies of the bioinspired architectures may be further exploited for developing new bioinspired metallic materials. The current theoretical analysis and processing route could offer means for designing and constructing the architectures in a more precise and efficient manner.

## Methods

**Bioinspired architectural design**. The three types of architectures reproduced here are all composed of relatively stiff and compliant phases in their natural prototypes in biological materials. However, they are distinguished by markedly different spatial arrangement of constituents, with representative examples shown in Fig. 1. The brick-and-mortar architecture features a staggered stacking of layers containing near hexagonal-shaped stiff platelets embedded within a compliant matrix (Fig. 1a)[35,36]. The Bouligand architecture is characterized by a periodically helicoidal twisting arrangement of constituent layers which comprise stiff fibers within the compliant matrix (Fig. 1b)[37,38]; these fibers are unidirectionally aligned within each layer but are twisted between adjacent layers by an identical angle between adjacent layers. The crossed-lamellar architecture also exhibits a layered arrangement but demonstrates a hierarchical nature encompassing at least two levels of length scales (Fig. 1c)[39,40]. The constituent layer is composed of a series of lamellae which are alternately stacked with each of the lamellae comprising well-aligned stiff platelets within the compliant matrix. However, these platelets display different orientations of alignment between adjacent lamellae, e.g., with the inclination angle of them ranging from 90° to 150° in the bivalve shell *Saxidomus purpuratus*[39,40].

Stereolithographic 3-D models of the above architectures were established using the Pro/Engineer 5.0 software (Parametric Technology Corporation, USA) as the templates for guiding the subsequent 3-D printing of the Ti-6Al-4V scaffolds. In this process, these architectures were idealized for simplification by ignoring the structural irregularities or imperfections in their natural prototypes[43]; nevertheless, the key features of the structural arrangement were retained in the models. These models were additionally regulated with appropriate modifications as per the technical requirements for 3-D printing, e.g., in order to remove the extra metal powders from the scaffolds after printing. The 3-D models and detailed structural characteristics are presented in Supplementary Fig. 1.

Specifically, for the brick-and-mortar architecture (Supplementary Fig. 1a), the constituent layers were composed of hexagonal-shaped units with an edge length of 2 mm in the Ti-6Al-4V scaffolds. These units were designed to be hollow with a wall thickness of 0.5 mm. Such design is to facilitate the removal of extra metal powders from the scaffolds through the hollow centers after printing—otherwise these powders cannot be removed through the narrow gaps between units. The units were stacked between layers in such a fashion that the center of the hexagon at each layer coincides well to the vertexes of hexagons at adjacent layers, i.e., with one-third of area overlapping between units. Consequently, the Ti-6Al-4V hexagon units with their hollow centers filled with Mg play a role as the bricks after melt infiltration of the 3-D printed scaffolds. This is different from the case in natural nacre where the bricks are dense, but still reproduce the main features of its spatial arrangement. Such a design, however, does lead to a constitution of stiff and soft phases in the brick-and-mortar structure for comparison to the other two architectures under study.

For the Bouligand architecture (Supplementary Fig. 1b), the Ti-6Al-4V fibers, with a diameter of 0.5 mm and aligned in parallel with an interspacing of 2 mm within each layer, were twisted between layers in a clockwise fashion from the bottom-up. The pitch or twisting angle was set as 15° to be qualitatively consistent with that in the dactyl club of *Odontodactylus scyllarus* (~15°)[37,38].

For the crossed-lamellar architecture (Supplementary Fig. 1c), rectangular-shaped Ti-6Al-4V platelets with a dimension of 2 × 1 × 0.5 mm and an interspacing of 1.3 mm were assembled into a series of lamellae with the same thickness of 1 mm. These platelets were inclined by ±50° with respect to the vertical direction (*y* axis) of the model, leading to a width of lamellae of 1.53 mm. A symmetrical arrangement was designed for the platelets between adjacent lamellae along their boundaries, leading to an inclination angle of these platelets of 100° which resembles that in the *Saxidomus purpuratus* shell (90–150°)[39,40].

The finest structural dimensions, i.e., the wall thickness for the brick-and-mortar architecture, the fiber diameter for the Bouligand architecture, and the platelet

thickness for the crossed-lamellar architecture, were all set to be 0.5 mm in the above models. This was determined by the diameter of the electron beam (~0.2 mm) of the 3-D printer which created a melt pool with a width of ~0.4 mm and a depth of ~0.3 mm in the printing process. The constituent layers were bonded together by the fusion joints at their contacts by 3-D printing, ensuring good continuity and integrity of the scaffolds. These interconnections are reminiscent of the similar designs in their natural prototypes, such as the existence of mineral bridges between platelets in the brick-and-mortar and crossed-lamellar architectures in mollusk shells and the fibers interspersing through different layers in the Bouligand architecture[35–38,65,66]. The models were all designed into a cuboid shape with dimensions of 70 × 70 × 12 mm. This ensured the containment of a large number of layers along their thickness direction, i.e., 24 layers for the brick-and-mortar and Bouligand architectures and 12 layers for the crossed-lamellar architecture, so as to fully embody the effects of the bioinspired architectures in subsequent mechanical testing. These models were then sliced along the bottom-up direction using the Materialise Magics 21.0 software (Materialise, Belgium) into layers with a thickness of 50 μm such that they can be recognized to guide the subsequent 3-D printing process.

**3-D printing**. The sliced layers of the 3-D models were printed sequentially by selective electron beam melting using an Arcam A1 EBM equipment (Arcam AB, Sweden). The electron beam had a diameter of ~200 μm and was subjected to a constant accelerating voltage of 60 kV. Gas-atomized Ti-6Al-4V powders (Arcam AB, Sweden) with nearly spherical geometry and an average diameter of ~77 μm were used for printing. To avoid oxidation of the melt while reducing obstruction of the electron beam, the printing process was conducted in a vacuum of ~10⁻³ mbar which was regulated using helium gas. The thickness of each powder layer was set at 50 μm, which was consistent with that of the sliced layers in the digital models. Before melting, the powder layer was pre-heated to ~730 °C via electron beam pre-scanning for 11 times at a high scanning speed of 15 ms⁻¹ with a beam current of 30 mA. This functions to minimize the thermal shrinkage and facilitate the fusion between powders during the melting process[67]. Then, the powders were selectively melted in accordance with the sliced layers of models with the electron beam at a scanning speed of 0.5 ms⁻¹ with a beam current of 2.4 mA. After printing, the extra powders were removed from the printed scaffolds using high-speed argon flow under a pressure of ~0.5 MPa. To promote further densification and reduce the internal residual stress, the printed Ti-6Al-4V scaffolds were subsequently heat-treated at 900 °C for 2 h in flowing argon gas using a graphite resistance furnace. As a comparison, dense Ti-6Al-4V alloy was also printed and heat-treated fully in line with the above procedures.

**Melt infiltration**. Blocks of pure Mg produced by the silicothermic reduction method, i.e., via thermal reduction of magnesium oxide with silicon[47], were placed on top of the 3-D printed Ti-6Al-4V scaffolds in a high-purity graphite crucible. The materials were heated in flowing argon gas to 800 °C, i.e., ~150 °C higher than the melting point of Mg, at a heating rate of 10 °C min⁻¹, held for 10 min, and then slowly cooled in the furnace. Before infiltration, the Mg blocks were mechanically burnished, prior to being ultrasonically cleaned along with the scaffolds in acetone to reduce contamination. Combinations of dense Ti-6Al-4V alloy and Mg bonded by a straight interface were also prepared following the same procedures.

**Microstructural characterization**. The densities of the infiltrated composites were measured using the Archimedes method[68]. Optical imaging was performed using an Axio Observer Z1 microscope (Zeiss, Germany). Before imaging, the composites were ground and polished to a surface finish of ~0.5 μm and then etched by immersing in an aqueous solution of 1.07 wt.% $HNO_3$, 0.68 wt.% $CH_3COOH$, and 0.65 wt.% $H_2C_2O_4$ for ~8 s. Scanning electron microscopy (SEM) imaging was conducted using an Inspect F50 field-emission scanning electron microscope (FEI, USA) operating at an accelerating voltage of 20 kV. The distribution of elements was measured by energy-dispersive X-ray spectroscopy (EDS) using an Oxford Model 7426 spectrometer (Oxford Instruments, UK). Local phase constitutions were determined by micro-zone X-ray diffraction (XRD) using a Bruker D8 Discover X-ray micro-diffractometer (Bruker AXS, Germany) with Co-Kα radiation operating at an accelerating voltage of 40 kV. The diameter of the X-ray beam was ~50 μm.

The 3-D architectures of the composites were characterized by X-ray diffraction topography (XRT) imaging using an Xradia Versa XRM-500 3-D X-ray microscope (Xradia, USA) operating at an accelerating voltage of 80 kV. The spatial distribution of cracks within the samples after mechanical testing was visualized by X-ray computed tomography (CT) using a YCT Modular non-destructive testing system (YXLON, Germany) operating at an accelerating voltage of 160 kV. CT imaging with higher-energy X-rays than XRT was employed here because the specimens for mechanical testing had much larger dimensions than those for microstructural characterization. For both XRT and CT measurements, the samples were rotated by 360° around the normal axis of the X-ray source and detector and imaged at equal intervals. A total of 1600 and 900 slices of 2D projections were acquired for the XRT and CT imaging, respectively. These projections were reconstructed to 3-D volume renderings based on the Fourier back-projection algorithm. The spatial resolutions of the 3-D XRT and CT images

were ~12 μm and ~20 μm per pixel, respectively. All the images were processed and analyzed using the Avizo Fire 9.5 software (Visualization Sciences Group, France).

**Residual stress measurements**. The residual stresses in the infiltrated composites were measured using the X-ray diffraction technique with an LXRD residual stress analyzer (Proto, Canada). The residual stress $\sigma_R$ in an individual phase can be accessed as follows:

$$\sigma_R = -\frac{\pi E \cos\theta_0}{360(1+\nu)}\frac{\partial 2\theta}{\partial \sin^2\Psi}, \tag{1}$$

where $E$ and $\nu$ are the Young's modulus and Poisson's ratio. $\theta_0$ is the X-ray diffraction angle for a specific crystal plane without residual stress, which is respectively 140.882° for Mg (303) plane under Cu-Kα radiation and 142.041° for Ti (211) plane under Co-Kα radiation. $\Psi$ is the incidence angle of the X-rays which was set to be 0°, 15°, 30°, and 45°. $\theta$ is the diffraction angle corresponding to each $\Psi$ at the presence of residual stress[69].

**Quasi-static uniaxial tensile testing**. Quasi-static uniaxial tensile tests were performed at room temperature using an Instron 5582 testing machine (Instron Corp., USA) at a fixed strain rate of $10^{-3}$ s$^{-1}$ with the strain monitored using an Instron 2620-601 extensometer. Dog-bone-shaped tensile specimens with a gauge length of 18 mm and cross-section of $12 \times 8$ mm were extracted from the infiltrated composites by electrical discharge machining, and then mechanically ground and polished to ~1 μm surface finish before testing. The tensile loads were applied along the $x$ axes in the $x$-$y$ planes for the bioinspired architectures, i.e., with the loading direction parallel to the constituent layers for all the composites, as illustrated in Fig. 1 and Supplementary Fig. 1. Such a loading configuration is qualitatively consistent with the situation in their biological prototypes where the resultant tensile stresses are principally in the normal plane of their thickness direction under service loading conditions[35–40]. Specifically, the tensile directions were respectively (i) along the longest diagonal of the hexagons for the brick-and-mortar architecture, (ii) parallel to the fibers at the outermost layers for the Bouligand architecture, and (iii) perpendicular to the boundaries between lamellae for the crossed-lamellar architecture. These loading configurations are clearly illustrated in Supplementary Fig. 2. At least three samples were tested for each set of composites with specific architectures; the results are presented in form of mean ± standard deviation. The differences in tensile properties among the three bioinspired composites were tested using a two-tailed Student's $t$ test, with $P < 0.05$ being considered as statistically significant. The tensile properties of the 3-D printed Ti-6Al-4V scaffolds without infiltration of Mg were also tested fully in line with the above procedures. Additionally, uniaxial tensile tests were performed on pure Mg, which was solidified along with the composites and the 3-D printed dense Ti-6Al-4V alloy for comparison. Further, the interfacial bonding strengths for combinations of these materials, bonded by a straight interface, were tensile tested with the interface aligned perpendicular to the tensile loading direction.

**Fracture toughness measurements**. The fracture toughness was measured for the composites and 3-D printed dense Ti-6Al-4V alloy using compact-tension C(T) specimens in accordance with the standard ASTM E1820[59]. The C(T) samples had a width $W$ of 40 mm and thickness $B$ of 10 mm and were loaded by two pins with a diameter of 0.24 $W$ inserted through the loading holes. Prior to testing, the samples were fatigue pre-cracked to a crack length $a_0$ of ~0.45 $W$, using an electro-servo MTS Lanmark 370.25 testing machine (MTS Corp., USA). This was operated under a load control at $\Delta K = 0.5$–2.0 MPa m$^{0.5}$ using a load ratio (the ratio of the minimum to the maximum loads) of 0.06 at 10 Hz frequency. The cracking planes were perpendicular to the constituent layers for all the three bioinspired architectures, i.e., parallel to the $z$ axes as illustrated in Fig. 1 and Supplementary Figs. 1 and 2. The loading directions for fracture toughness tests conformed to those for uniaxial tensile testing for the Bouligand and crossed-lamellar architectures, but were perpendicular to that for the brick-and-mortar architecture in view of the fact that it is nearly isotropic in-plane. As such, the directions for resultant ideally mode I crack extension were respectively (i) parallel to the longest diagonal of the hexagons for the brick-and-mortar architecture (in the $x$-$z$ plane), (ii) perpendicular to the fibers at the outermost layers for the Bouligand architecture (in the $y$-$z$ plane), and (iii) parallel to the boundaries between lamellae for the crossed-lamellar architecture (in the $y$-$z$ plane). The loading configurations are also illustrated in Supplementary Fig. 2.

Nonlinear elastic-fracture mechanics-based R-curve measurements were employed to evaluate the fracture toughness. This method can not only quantify the material's resistance to the onset of crack extension, but also to subsequent crack growth. The samples were loaded in uniaxial tension using the MTS Lanmark 370.25 machine under a displacement control mode at a fixed stress intensity rate of 1 MPa m$^{0.5}$ s$^{-1}$. The crack growth was determined using the single-specimen method where repeated unloading/reloading sequences were applied to produce stepwise crack extension[59]. The crack lengths were calculated from the elastic compliance measured during the unloading cycles.

To establish the R-curve for a given specimen, the $J$-integral corresponding to a specific crack length $a_{(i)}$ at the $i$th unloading/reloading cycle can be divided into the elastic component $J_{el(i)}$ and plastic component $J_{pl(i)}$ as $J_{(i)} = J_{el(i)} + J_{pl(i)}$[59].

The elastic component can be obtained from the elastic stress intensity $K_{(i)}$ in the plane-strain condition as:

$$J_{el(i)} = \frac{K_{(i)}^2(1-\nu^2)}{E}, \tag{2}$$

where $E$ is Young's modulus of the composite which can be determined by tensile testing and $\nu$ is the Poisson's ratio which was set as 0.345 (the Poisson's ratios were 0.34 and 0.35, respectively, for the Ti-6Al-4V alloy and pure Mg[18,19]). $K_{(i)}$ can be determined according to the following relationship:

$$K_{(i)} = \frac{P_{(i)}}{B\sqrt{W}}f\left(\frac{a_{(i)}}{W}\right), \tag{3}$$

with

$$f\left(\frac{a_{(i)}}{W}\right) = \frac{\left(2 + \frac{a_{(i)}}{W}\right)\left[0.886 + 4.64\frac{a_{(i)}}{W} - 13.32\left(\frac{a_{(i)}}{W}\right)^2 + 14.72\left(\frac{a_{(i)}}{W}\right)^3 - 5.6\left(\frac{a_{(i)}}{W}\right)^4\right]}{\left(1 - \frac{a_{(i)}}{W}\right)^{\frac{3}{2}}}, \tag{4}$$

where $P_{(i)}$ is the applied load prior to the $i$th unloading. The plastic component $J_{pl(i)}$ can be calculated in an accumulative fashion from that of the previous unloading/reloading cycle $J_{pl(i-1)}$ using:

$$J_{pl(i)} = \left[J_{pl(i-1)} + \frac{[2 + 0.522(1 - a_{(i-1)}/W)]\Delta A_{pl(i)}}{B(W - a_{(i-1)})}\right]\left[1 - \frac{(a_{(i)} - a_{(i-1)})[1 + 0.76(1 - a_{(i-1)}/W)]}{W - a_{(i-1)}}\right], \tag{5}$$

where $\Delta A_{pl(i)}$ is the increment of plastic area under the load versus load-line displacement curve for $i$th unloading/reloading cycle. The obtained $J$-integral values were finally corrected and examined in terms of their validity in strict accordance with the standard ASTM E1820[59].

**Instrumented Charpy impact testing**. Instrumented Charpy impact tests were conducted at room temperature using a ZBC2452-C pendulum impact testing machine (SANS, China) equipped with an impact striker with a blade radius of 2 mm. The impact speed and impact energy of the striker were 5.24 ms$^{-1}$ and 450 J, respectively. Unnotched cuboid samples with a length of 55 mm and a cross-section of $10 \times 10$ mm were used for impact tests with the loading span set as 40 mm, in accordance with the ISO 148-1:2006 Standard[70]. Before testing, all the samples were ground and polished to ~1 μm surface finish. The impact loads were maintained normal to the constituent layers for all the three types of bioinspired composites, but along different directions with respect to their architectures, as illustrated in Supplementary Fig. 2. Specifically, the loading directions were along the $x$, $y$, and $z$ axes for the crossed-lamellar architecture, but due to their nearly isotropic in-plane nature (i.e., in the $x$-$y$ plane), only loading along the $y$ and $z$ axes was examined for the brick-and-mortar and Bouligand architectures. At least three samples were tested for each loading configuration of the bioinspired composites. The impact toughness $\alpha_\kappa$, in terms of the absorption of impact energy per unit cross-sectional area, is presented in form of mean ± standard deviation. The statistical differences among these groups were tested using two-tailed Student's $t$ test with a significance level of $P$ value < 0.05.

## Data availability

The data that support the findings of this study are available in Materials Cloud with the https://doi.org/10.6084/m9.figshare.19646721, and are also available from the corresponding author upon reasonable request.

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

## Acknowledgements

The authors are grateful for the financial support by the National Key R&D Program of China under grant number 2020YFA0710404, the National Natural Science Foundation of China under grant numbers 51871216 and 52173269, the KC Wong Education Foundation under grant number GJTD-2020-09, the State Key Laboratory for Modification of Chemical Fibers and Polymer Materials at Donghua University, the Opening Project of National Key Laboratory of Shock Wave and Detonation Physics under grant number 6142A03203002, and the Youth Innovation Promotion Association CAS. ROR was supported by the Multi-University Research Initiative under grant number AFOSR-FA9550-15-1-0009 from the Air Force Office of Scientific Research.

## Author contributions

Z.L., Q.Y., and Z.Z. designed the research; M.Z. and J.Z. fabricated the composites; S.L. and D.R. carried out 3-D printing; M.Z. and N.Z. characterized the microstructures; M.Z., N.Z., and R.Q. measured the mechanical properties; M.Z., Z.L., Q.Y., F.B., Z.Z., and R.O.R. analyzed and discussed the data; M.Z. and Z.L. wrote the initial paper; Z.Z., F.B., and R.O.R. revised the paper.

## Competing interests

The authors declare no competing interests.
