## [Peer Review File · Nature Communications]

Title: On the damage tolerance of 3-D printed Mg-Ti interpenetrating-phase composites with bioinspired architecturesREVIEWER COMMENTS

Reviewer #1 (Remarks to the Author):

This paper investigates the mechanical properties of the bioinspired 3D printed interpenetrating-phase Mg-Ti composites. The topic is interesting. It is publishable after some details are cleared. There are some comments for the authors to consider.

Damage tolerant 3D printed interpenetrating-phase composites have been studied since 10 years ago. Relevant references should be reviewed and compared, such as

L. F. Wang, J. Lau, E. L. Thomas, and M. C. Boyce, "Co-Continuous Composite Materials for Stiffness, Strength and Energy Dissipation", *Advanced Materials* 23, 1524-1529 (2011).

J. H. Lee, L. F. Wang, S. Kooi, M. C. Boyce and E. L. Thomas, "Periodic Bicontinuous Composites for High Specific Energy Absorption", *Nano Letters* 12, 4392-4396 (2012).

T. T. Li, Y. Y. Chen, and L. F. Wang, "Enhanced Fracture Toughness in Architected Interpenetrating Phase Composites by 3D Printing", *Composites Science and Technology* 167, 251-259 (2018).

For the first brick-and-mortar architecture mimicking nacre, the volume fractions of the Ti-6Al-4V phase is ~37.6% as matrix (which is stiffer), which is quite different from nacre composition. In nacre, matrix is softer, and the volume fraction is only 3-5%. Is there any particular reason to do so?

Table 1 should also list the mechanical properties of Ti-6Al-4V, as well as in Figure 3a. It's not clear how these results are compared to those for Ti-6Al-4V.

The second step of pressureless infiltration of the scaffolds with pure Mg melt may be tricky and can raise a lot of questions about the interface.

Except for the density measurement, is there any evidence showing the absence of pores in the composites?

How to determine the interfacial properties between the Ti-6Al-4V and the Mg? Are there any weaknesses in the interface? It would be helpful to have some experimental tests to verify the assumptions, similar to F. Liu, T. T. Li, X. H. Jiang, Z. A. Jia, Z. P. Xu and L. F. Wang, "The Effect of Material Mixing on Interfacial Stiffness and Strength of Multi-material Additive Manufacturing", *Additive Manufacturing* 36, 101502 (2020).

What are the fracture toughness for Pure Mg and Ti-6Al-4V respectively? It would be interesting to see all the results in Table 1 for direct comparison.

The bioinspired composites exhibited "Γ"-shaped R-curves, and how does it compare to results from Reference 7?

Reviewer #2 (Remarks to the Author):

This work introduces a method for fabrication of metallic composite based on the 3D printed metallic scaffold with a high melting point and pressureless infiltrating melted metal. This research investigates different bioinspired architectures (brick and mortar, crossed-lamellar, and Bouligand) and offers the realization of tough, strong, and fracture/impact-resisting metallic composites.

The text is very well organized and well written. Although the idea of the bioinspired architected composite has already been reported in the literature, the fabrication of the metallic composite with arbitrary configurations is novel. This method can be used to fabricate any metallic composite with different melting points and complex microstructures. The modified theory is effective to evaluate important properties of this kind of composite. The experimental results are comprehensive and validate the authors' claims and theoretical studies. Please find my comments as follow:

(Comment 1)

The brick and mortar sample is not really representing the brick and mortar structures in nature. In nature, brick is in the form of strong hexagonal plates with softer flexible mortar. Here strong material (Ti alloy) roles as matrix and Mg works as hexagon plates. It will be similar if plates are made of Ti and the matrix is made of Mg. To make the scaffold feasible in 3D printing, researchers can consider some small connectivity parts between plates. A simpler solution can be rewording brick and mortar.

(Comment 2)

This is a composite material, and it will be a complete comparison if the results of pure scaffold are also tested, especially since the 30-50% weight ratio for scaffold is relatively high. Furthermore, the mechanical properties of Titanium, fabricated by this method, can be added. It is not clear if the material properties of scaffold will be affected by using the melt matrix or not. The authors should elaborate on the possibility of the generation of residual (thermal) stress in the produced samples.

(Comment 3)

Generally in composite material, the strength of the structure is being maximized when the direction of the microstructure is aligned with the loading direction. In Figure 4-b, " α " is the angle of two ligaments, and in $\alpha=0$ the whole structure will be pure Ti alloy (this may contradict logically with composite definition). Hence maximum strength happens in $\alpha=0$. In Figure 4-c, for $\beta=0$, the fibers are aligned with the loading direction and maximum strength should be occurred for $\beta=0$. Figure 4-b and 4-c seems to be incorrect, while Figure 4-d confirms that when Ti microstructures are aligned with loading direction the strength is maximized.

(Comment 4)

In Figures 4-b, 4-c and 4-d, the direction of the loading is not depicted on microstructures, making it hard to justify the results.

(Comment 5)

In Figure 4-c, diagrams of the composite have wavy behavior. Please explain the nature of this wavy

behavior and the rationale of the existence of pick points.

(Comment 6)

These are composite materials and it needs to show the mechanical properties with respect to the load direction.

(Comment 7)

In these kinds of composite, the weight or volume fraction of the two materials is an important parameter. It would be complete if the role of the weight fraction of two materials is studied for one structure. Is the improvement found by implementing specific bio-inspired design valid for alternative weight ratio or volume fraction of scaffold and matrix? In engineered composites, both architecture as well as weight/volume fraction are simultaneously optimized to achieve the performance improvement.

(Comment 8) (Fracture) toughness has been measured based on the standard pre-cracked test on bio-inspired samples. If toughness is measured directly from stress-strain curve or load-displacement in three-point bending as well as tensile test modes, the authors should elaborate in supplementary or main text how the toughness measurements and the performance of the bioinspired composites (presented in Figure 7) would vary from the reported fracture toughness results.

Response to Reviewers

We sincerely thank both the reviewers for their very constructive and comprehensive review of our manuscript. We found their advice and suggestions to be particularly useful and have attempted to revise the paper exactly along the lines that they suggest. Our revisions are indicated by a red font in the text. We believe that the manuscript is distinctly improved by these revisions. We hope that you concur. Our specific point-to-point responses to the reviewers' comments are described below:

Reviewer #1

This paper investigates the mechanical properties of the bioinspired 3D printed interpenetrating-phase Mg-Ti composites. The topic is interesting. It is publishable after some details are cleared. There are some comments for the authors to consider.

Response: We appreciate the encouraging comment and positive evaluation.

Comment 1: *Damage tolerant 3D printed interpenetrating-phase composites have been studied since 10 years ago. Relevant references should be reviewed and compared, such as L. F. Wang, J. Lau, E. L. Thomas, and M. C. Boyce, "Co-Continuous Composite Materials for Stiffness, Strength and Energy Dissipation", *Advanced Materials* 23, 1524-1529 (2011). J. H. Lee, L. F. Wang, S. Kooi, M. C. Boyce and E. L. Thomas, "Periodic Bicontinuous Composites for High Specific Energy Absorption", *Nano Letters* 12, 4392-4396 (2012). T. T. Li, Y. Y. Chen, and L. F. Wang, "Enhanced Fracture Toughness in Architected Interpenetrating Phase Composites by 3D Printing", *Composites Science and Technology* 167, 251-259 (2018).*

Response: We thank you for the valuable comment. Relevant references, which mainly deal with polymeric materials, have been cited and briefly introduced in the introduction section of revised manuscript. A comparison has also been made between these results with our study.

Comment 2: *For the first brick-and-mortar architecture mimicking nacre, the volume fractions of the Ti-6Al-4V phase is ~37.6% as matrix (which is stiffer), which is quite*

different from nacre composition. In nacre, matrix is softer, and the volume fraction is only 3-5%. Is there any particular reason to do so?

Response: We appreciate the valuable comment. The brick-and-mortar architecture was designed to be different from that of natural nacre because it is difficult to fabricate porous scaffolds of metals with a low porosity using the selective electron beam melting technique. If dense Ti-6Al-4V bricks were designed and printed into a scaffold with low porosity, the interspaces between adjacent bricks would become quite narrow and tortuous such that the extra metal powders cannot be completely removed through them from the scaffold after 3-D printing. To address this problem, the hexagonal-shaped Ti-6Al-4V units with an edge length of 2 mm were designed to be hollow with a wall thickness of 0.5 mm. This creates a hexagonal-shaped hollow center within each unit with an edge length of 1.5 mm, thereby allowing for the removal of extra metal powders through them. After melt infiltration of the scaffold, the hexagonal-shaped Ti-6Al-4V units with the internal magnesium which filled the hollow center can be regarded as the “bricks” like in natural nacre. As such, the volume fractions of stiff and soft phases and the detailed architectural characteristics of the bioinspired brick-and-mortar architecture are different from those of nacre. Nevertheless, the spatial configuration of architecture reproduces the designs of nacre in the following aspects: (1) the hexagonal-shapes of structural units, (2) the staggered stacking of units and the presence of interconnections between adjacent layers, and (3) the mutual interpenetration of stiff and soft phases in 3-D space. These characteristics were found to play a similar role as in nacre in toughening the composites. Such a design also leads to a similar constitution of stiff and soft phases in the brick-and-mortar architecture as for the other two architectures to assist comparison. Thanks to your valuable comment, we have attempted in the revised manuscript to more clearly describe the design of our bioinspired brick-and-mortar architecture as well as the differences and similarities with that of nacre.

Comment 3: *Table 1 should also list the mechanical properties of Ti-6Al-4V, as well as in Figure 3a. It's not clear how these results are compared to those for Ti-6Al-4V.*

Response: We appreciate the useful comment and good suggestion. The mechanical properties of 3-D printed dense samples and porous scaffolds of Ti-6Al-4V alloy, which were printed using the same parameters as for the composites, have been measured experimentally. The results have been added into Table 1 in the revised manuscript. A comparison has been made between the composites and Ti-6Al-4V alloy. As the strength of dense Ti-6Al-4V alloy far exceeds those of the scaffolds, Mg and the composites, it would become difficult to discern the stress-strain curves of the lower strength samples if they are plotted together with those of Ti-6Al-4V alloy in the same figure. Therefore, the tensile stress-strain curves for the Ti-6Al-4V alloy, of both dense blocks and porous scaffolds, have been presented in Supplementary Figures 7 in the revised manuscript (also shown here in Figure S1).

Figure S1. Tensile stress-strain curves of the (a) dense blocks and (b) porous scaffolds with different architectures for the 3-D printed Ti-6Al-4V alloy. The insets show the dog-bone shaped tensile specimens of them.

Comment 4: *The second step of pressureless infiltration of the scaffolds with pure Mg melt may be tricky and can raise a lot of questions about the interface. Except for the density measurement, is there any evidence showing the absence of pores in the composites?*

Response: We appreciate the careful review and valuable comment. In general, metal melts demonstrate a good wettability with metals owing to the good affinity under the action of strong metallic bonds, mutual solubility and even formation of intermetallic

compounds between them [1,2], making it possible to fill metal scaffolds with metal melts by pressureless infiltration. This has been experimentally verified for the system of titanium and magnesium [3-6]. A tight bonding can be formed between the two phases with no obvious pores present at the interfaces. Except for the density measurement, we have also characterized the structures of the bioinspired composites using scanning electron microscopy imaging by up to 5000 times magnification (Figures 2 and 6), X-ray diffraction topography imaging with a spatial resolution of ~12 μm per pixel (Figure 1), and X-ray computed tomography imaging with a spatial resolution of ~20 μm per pixel (Figures 3 and 6). No obvious pores were detected within the resolution ranges of these techniques. This, along with the density measurement, offers the experimental evidence showing that there were no obvious pores in the composites by pressureless infiltration.

Comment 5: *How to determine the interfacial properties between the Ti-6Al-4V and the Mg? Are there any weaknesses in the interface? It would be helpful to have some experimental tests to verify the assumptions, similar to F. Liu, T. T. Li, X. H. Jiang, Z. A. Jia, Z. P. Xu and L. F. Wang, "The Effect of Material Mixing on Interfacial Stiffness and Strength of Multi-material Additive Manufacturing", Additive Manufacturing 36, 101502 (2020).*

Response: We appreciate the valuable comment and useful suggestion. As evidenced by the energy-dispersive X-ray spectroscopy and micro-zone X-ray diffraction results (Figure 2), Si-containing intermetallic phases were precipitated at the interfaces between Ti-6Al-4V and magnesium phases. These precipitates may act as the prime sites for cracking upon loading, which is similar as the case observed in Mg-NiTi system [7,8]. In line with the good suggestion, we have attempted to determine the interfacial strength using uniaxial tension tests. As shown in Figure S2, a combination of Ti-6Al-4V and magnesium phases bonded by a straight interface at the gauge section, which was perpendicular to the tensile direction, was loaded by uniaxial tension until fracture. The ultimate tensile strength of the combination was measured to be around 58 MPa, approximating that of pure magnesium. Additionally, the fracture occurred in

the magnesium phase rather than along the interface, indicating that the interfacial strength was higher than that of magnesium.

Figure S2. Tensile stress-strain curve of the combination of Ti-6Al-4V and magnesium phases bonded by a straight interface at the gauge section. The inset shows the overall appearances of samples before and after tension.

On the other hand, as indicated by the literature suggested by the reviewer, the interfacial strength of composite is tightly associated with the materials mixing at interface and the roughness of interface. As shown in Figure S3, the Ti-6Al-4V and magnesium phases demonstrate a rough interface (as indicated by the dashed curve) and show materials mixing over a range of $\sim 100 \mu\text{m}$, as evidenced by the gradual changes in the contents of Mg and Ti elements. These characteristics are similar with those reported in literature. Additionally, as the thicknesses of Ti-6Al-4V and magnesium phases (hundreds of micrometers to several millimeters) significantly exceed that of the materials mixing region, the interfacial effects should be much less significant in the composites as compared to their spatial architectures. The structure and strength of interface have been described in more detail in the revised manuscript with the experimental results presented in Supplementary Figures 5 and 6. Relevant reference has also been cited.

Figure S3. Micrograph of the interface between Ti-6Al-4V and magnesium phases and the energy dispersive X-ray spectroscopy results showing the linear distributions of elements across the interface (along the dashed black line).

Comment 6: *What are the fracture toughness for pure Mg and Ti-6Al-4V respectively? It would be interesting to see all the results in Table 1 for direct comparison. The bioinspired composites exhibited “I”-shaped R-curves, and how does it compare to results from Reference 7?*

Response: We thank you for the useful comment and good suggestion. We have measured the fracture toughness of 3-D printed Ti-6Al-4V alloy ($\sim 79.6 \text{ MPa m}^{1/2}$) using compact-tension specimens, which is the same as for the composites, in line with the standard ASTM E1820. The fracture toughness of pure magnesium has been reported in literature to be in a range of 6-20 $\text{MPa m}^{1/2}$, which is associated with the processing condition, *e.g.*, of as-cast or extruded states. These results have been added into Table 1 in the revised manuscript for direct comparison. The measured R-curves of Ti-6Al-4V alloy have also been presented in Supplementary Figure 12 (also shown here in Figure S4). The R-curve behavior of our bioinspired composites has been compared with the results from ref. 7. Specifically, the *J*-based R-curves for all the composites, both in this study and from ref. 7, show a continuous rising trend with crack extension,

although the toughness values of the polymeric materials in ref. 7 are markedly lower than those of the metal composites in our study. Our composites all exhibit “T”-shaped R-curves, which is similar with the composites with brick-and-mortar and crossed-lamellar architectures in ref. 7. However, the composite with rotating plywood architecture (*i.e.*, Bouligand architecture) displays “J”-shaped R-curves in ref. 7. Such behavior has been attributed to the fact that the polymeric composite exhibits relatively low resistance to the onset of crack extension (much lower than metals), but becomes increasingly more resistant to crack growth owing to the notable crack deflection and twisting along its architecture – the crack tip can be obviously deviated from type I stress state.

Figure S4. Variations in the J -integral as a function of the crack extension Δa along with the fitted R-curve for the 3-D printed dense Ti-6Al-4V alloy. The dash line for determining the fracture toughness has a slope of $(\sigma_{YS} + \sigma_{UTS})$ and corresponds to a 0.2 mm offset strain. The inset shows the sample for compact tension testing.

Reviewer #2

This work introduces a method for fabrication of metallic composite based on the 3D printed metallic scaffold with a high melting point and pressureless infiltrating melted

metal. This research investigates different bioinspired architectures (brick and mortar, crossed-lamellar, and Bouligand) and offers the realization of tough, strong, and fracture/impact-resisting metallic composites. The text is very well organized and well written. Although the idea of the bioinspired architected composite has already been reported in the literature, the fabrication of the metallic composite with arbitrary configurations is novel. This method can be used to fabricate any metallic composite with different melting points and complex microstructures. The modified theory is effective to evaluate important properties of this kind of composite. The experimental results are comprehensive and validate the authors' claims and theoretical studies. Please find my comments as follow.

Response: We appreciate the encouraging comment and positive evaluation.

Comment 1: *The brick and mortar sample is not really representing the brick and mortar structures in nature. In nature, brick is in the form of strong hexagonal plates with softer flexible mortar. Here strong material (Ti alloy) roles as matrix and Mg works as hexagon plates. It will be similar if plates are made of Ti and the matrix is made of Mg. To make the scaffold feasible in 3D printing, researchers can consider some small connectivity parts between plates. A simpler solution can be rewording brick and mortar.*

Response: We appreciate the valuable comment and good suggestion. We designed and printed dense Ti-6Al-4V plates connected by struts of 0.5 mm in diameter at our preliminary attempt. The main problem was that the plates between adjacent layers were overlapped (by one third of area) and were strongly interconnected by fusion points at the overlapped regions. This led to a narrow interspace between layers such that the extra metal powders could not be removed from the scaffold after printing. To address this problem, the hexagonal-shaped Ti-6Al-4V units with an edge length of 2 mm were designed to be hollow with a wall thickness of 0.5 mm. This creates a hexagonal-shaped hollow center within each unit with an edge length of 1.5 mm, thereby allowing for the removal of extra metal powders through them. After melt infiltration of the scaffold, the hexagonal-shaped Ti-6Al-4V units with the internal magnesium which filled the hollow center can be regarded as the “bricks” like in nacre.

The spatial configuration of architecture reproduces the designs of nacre in the following aspects: (1) the hexagonal-shapes of structural units, (2) the staggered stacking of units and the presence of interconnections between adjacent layers, and (3) the mutual interpenetration of stiff and soft phases in 3-D space. These characteristics were found to play a similar role as in nacre in toughening the composites. Such design also leads to a similar constitution of stiff and soft phases in the brick-and-mortar architecture as for the other two architectures to assist comparison. In line with the good suggestion by the reviewer, we will construct connections not only between plates within the same layer, but between adjacent layers to increase the interspace in our future work. An effort is required when increasing the interspace to avoid the collapse of layers due to the absence of fusion points while ensuring the removal of extra metal powders. This will be tried and reported in our future study but is not included in this paper. However, following your valuable comments, the design of the brick-and-mortar architecture as well as its differences and similarities with that of nacre has been described in more detail in the revised manuscript.

Comment 2: *This is a composite material, and it will be a complete comparison if the results of pure scaffold are also tested, especially since the 30-50% weight ratio for scaffold is relatively high. Furthermore, the mechanical properties of Titanium, fabricated by this method, can be added. It is not clear if the material properties of scaffold will be affected by using the melt matrix or not. The authors should elaborate on the possibility of the generation of residual (thermal) stress in the produced samples.*

Response: We appreciate the careful review and valuable comment. In line with the good suggestion, we have tested the tensile properties of pure scaffolds with different architectures and the Ti-6Al-4V alloy fabricated by the same 3-D printing method as for the composites (Supplementary Figures 7). The fracture toughness of dense Ti-6Al-4V alloy has also been measured and added into the revised manuscript (Supplementary Figure 12). A comparison has been made with these materials (as listed in Table 1). We have further characterized the microstructures and mechanical properties of the Ti-6Al-4V scaffold before and after melt infiltration. As shown in Figure S5a, the scaffold

exhibits similar microstructures, *i.e.*, of fine basket-weave texture composed of α -Ti and β -Ti phases, before and after melt infiltration. The load-displacement curves measured by nanoindentation testing along with the deduced hardness and elastic modulus are also almost identical for the scaffold before and after melt infiltration (Figure S5b). As such, the material properties of scaffold were nearly unaffected by melt infiltration. These results have also been added into the revised manuscript (Supplementary Figures 4).

Figure S5. Representative (a) microstructures and (b) load-displacement curves measured by nanoindentation testing along with the deduced hardness and elastic modulus for the Ti-6Al-4V scaffold before and after melt infiltration. The curve for the infiltrated sample was horizontally shifted for clarity.

Additionally, we have measured the residual stresses in both Ti-6Al-4V and magnesium phases after melt infiltration using X-ray diffraction technique according to the $\sin^2\Psi$ method (the residual stress normal to selected crystal plane can be determined by the offset angle of diffraction peak at different incidence angles Ψ) [9]. As shown in Figure S6, the residual stresses were measured to be \sim -5.4 MPa and \sim -2.4 MPa respectively for Mg (303) plane and Ti (211) plane in the composite. Such low residual stresses, which are lower than the strengths of the two phases by one to two orders of magnitude, may be associated with the slow furnace cooling after melt infiltration. These results have also been added into the revised manuscript (Supplementary Figures

3).

Figure S6. X-ray diffraction peaks at incidence angles Ψ of 0°, 15°, 30° and 45°, and the linear fitting results of $\partial 2\theta / \partial \sin^2\Psi$ in the 2θ - $\sin^2\Psi$ plots for (a) Mg (303) plane under Cu-K α radiation and (b) Ti (211) plane under Co-K α radiation in the infiltrated composite (the crossed-lamellar architecture was taken as an example).

Comment 3: Generally in composite material, the strength of the structure is being maximized when the direction of the microstructure is aligned with the loading direction. In Figure 4-b, “ α ” is the angle of two ligaments, and in $\alpha=0$ the whole structure will be pure Ti alloy (this may contradict logically with composite definition). Hence maximum strength happens in $\alpha=0$. In Figure 4-c, for $\beta=0$, the fibers are aligned with the loading direction and maximum strength should be occurred for $\beta=0$. Figure 4-b and 4-c seems to be incorrect, while Figure 4-d confirms that when Ti microstructures are aligned with loading direction the strength is maximized.

Response: We appreciate the careful review and valuable comment. We apologize that we made a mistake in the loading directions in Figure 4b and c. The tensile load was

applied along the transverse direction, instead of the longitudinal one, in the original version of figures. We have corrected this mistake and have clearly depicted the loading directions in the figure in the revised manuscript. It can be found that the theoretical results are fully consistent with those predicted by the reviewer – the maximum strengths occur at $\alpha=0$ and $\beta=0$ for the brick-and-mortar and Bouligand architectures, respectively. In addition, the phase constitution of the composite was fixed while calculation by fixing the interspaces between adjacent struts in element I for the brick-and-mortar architecture. As such, the variation in α will not change the contents of the two phases but only leads to a change in the length of element II. We appreciate the useful comment and have corrected the corresponding figures and text in the revised manuscript.

Comment 4: *In Figures 4-b, 4-c and 4-d, the direction of the loading is not depicted on microstructures, making it hard to justify the results.*

Response: We appreciate the good suggestion and have depicted the direction of the loading on microstructures in Figures 4b-d in the revised manuscript.

Comment 5: *In Figure 4-c, diagrams of the composite have wavy behavior. Please explain the nature of this wavy behavior and the rationale of the existence of pick points.*

Response: We appreciate the valuable comment and have explained the wavy behavior and the existence of peak points on the diagrams in the revised manuscript. In our laminate theory analysis, a laminate containing a total of 24 laminae through its thickness direction was considered for the Bouligand architecture, which was consistent with the experiments. As shown in Figure S7, the deviation angles of fibers in these laminae with respect to the loading direction show variations and fluctuations as β increases. These fibers may even be overlapped between different lamellae at specific angles of β . Specifically, the deviation of fibers initially increases with increasing β for all the lamellae, leading to a sharp decrease in the strength and modulus. This trend continues until $\beta=90^\circ/24=3.75^\circ$ where the fibers at the first layers show decreased deviation with increasing β . With the further increase of β , the fibers at different layers show different varying trends in their deviation, leading to variations in the overall

degree of deviation. This can be exemplified by the cases as β increases from 6° to 9° and then to 12° , where the overall deviation firstly increases and then decreases (Figure S7a). As such, the diagrams of the composite exhibit a wavy behavior.

Figure S7. Schematic illustrations of the orientations of fibers for the Bouligand architecture at specific angles of (a) $\beta=6^\circ, 9^\circ, 12^\circ$ (b) $\beta=29^\circ, 30^\circ, 31^\circ$, and (c) $\beta=35^\circ, 36^\circ, 37^\circ$. The fiber orientations in different laminae, as indicated by the serial numbers, are represented by the black lines. The black and red arrows indicate the loading direction and twisting direction of laminae, respectively.

Additionally, at specific angles that are divisible by 90° (such as $\beta=30^\circ$), the fibers in some of the layers become perpendicular to the loading direction and thereby play a

minimal role in stiffening/strengthening the composite (Figure S7b), leading to the minimum values on the diagrams. At specific angles especially those are divisible by 180° but not divisible by 90° (such as $\beta=36^\circ$), the fibers at some of the layers are aligned right along the loading direction and none of them are perpendicular to the loading direction (Figure S7c), leading to the peak points on the diagrams. The emergence of relatively sharp peaks for strengths, than for modulus, is caused by the fact that the yield and ultimate tensile strengths are approximated using the specific stresses corresponding respectively to the failure of the first and final laminae. When the fibers between different laminae are overlapped, the failure occurs simultaneously in two or more laminae and thereby needs higher stresses than for one individual lamina.

Comment 6: *These are composite materials and it needs to show the mechanical properties with respect to the load direction.*

Response: In line with the good suggestion, the loading direction has been depicted in the figures and clearly explained in the text in the revised manuscript.

Comment 7: *In these kinds of composite, the weight or volume fraction of the two materials is an important parameter. It would be complete if the role of the weight fraction of two materials is studied for one structure. Is the improvement found by implementing specific bio-inspired design valid for alternative weight ratio or volume fraction of scaffold and matrix? In engineered composites, both architecture as well as weight/volume fraction are simultaneously optimized to achieve the performance improvement.*

Response: We appreciate the useful suggestion and have investigated the effects of phase constitution (in volume fraction) on the Young's modulus and strengths for the three types of architectures based on our laminate theory modeling. Figure S8 shows the theoretical results for the variations in the Young's modulus as a function of the volume fraction of Ti-6Al-4V phase for the composites with different values of specific angles α , β and γ . The modulus increases monotonically with the increase of Ti-6Al-4V content for all the three types of architectures, with all the values ranging between the upper and lower bounds determined respectively by the Voigt and Reuss models in line

with the rule-of-mixtures. Figure S9 shows the theoretical results for the variations in the Young's modulus and strengths of the composites with different volume fractions of Ti-6Al-4V phase (by 20 vol.% addition and subtraction on the basis of experimental values) as a function of the specific angles. Although their absolute values are different, these mechanical properties demonstrate similar varying trends for specific architectures despite the differences in phase constitution. Specifically, the strengths of the brick-and-mortar architecture are comparable for different phase constitutions at the relatively large range of α because they are mainly dominated by the weak magnesium phase (failure is prone to occur in element II). A similar case occurs for the cross-lamellar architecture at the relatively small range of γ . These results have been added into the revised manuscript (Supplementary Figures 10 and 11). The effects of phase constitution on the mechanical properties have also been briefly elucidated.

Figure S8. Theoretical results for the variations in the Young's modulus as a function of the volume fraction of Ti-6Al-4V phase in the composites with bioinspired (a) brick-and-mortar, (b) Bouligand, and (c) crossed-lamellar architectures at fixed values of specific angles α , β and γ . The dashed curves indicate the upper and lower bounds determined respectively by the Voigt and Reuss models in line with the rule-of-mixtures.

Figure S9. Theoretical results for the variations in the Young's modulus, yield strength σ_{YS} and ultimate tensile strength σ_{UTS} as a function of the specific angles α , β and γ , for the composites with bioinspired (a) brick-and-mortar, (b) Bouligand, and (c) crossed-lamellar architectures. Three different volume fractions of Ti-6Al-4V phase (with 20 vol.% addition and subtraction on the basis of experimental data) were considered for each type of architecture.

Comment 8: *(Fracture) toughness has been measured based on the standard pre-cracked test on bio-inspired samples. If toughness is measured directly from stress-strain curve or load-displacement in three-point bending as well as tensile test modes, the authors should elaborate in supplementary or main text how the toughness measurements and the performance of the bioinspired composites (presented in Figure 7) would vary from the reported fracture toughness results.*

Response: We appreciate the valuable comment. The mode I fracture toughness can also be measured directly from the load-displacement in three-point bending or tensile test modes in accordance with the standard ASTM E399 [10,11]. However, such measurement can only quantify the toughness at the onset of crack extension, K_{Ic} , but cannot evaluate the material's resistance to subsequent crack growth – it is generally at this stage that the bioinspired architectures play a key toughening role. Therefore, we measured J -based resistance curves to evaluate the fracture toughness of bioinspired

composites in accordance with the standard ASTM E1820 [10,12]. The R-curves obtained by this method can not only reflect the resistance to crack growth, but to the onset of crack extension using the equivalent stress intensity K_{JIC} derived from J_{IC} . In line with the good suggestion by the reviewer, we have measured the fracture toughness K_{IC} directly from the load-displacement curves for the bioinspired composites (the sample geometry and testing method in this study also conform to the standard ASTM E399). These values have been used in the revised manuscript (Table 1 and Figure 7). K_{IC} values, of $\sim 22.40 \text{ MPa m}^{1/2}$, $\sim 25.04 \text{ MPa m}^{1/2}$ and $\sim 27.74 \text{ MPa m}^{1/2}$ respectively for the brick-and-mortar, Bouligand, and crossed-lamellar architectures, are slightly lower only by 1-2 $\text{MPa m}^{1/2}$ than K_{JIC} . Such a small difference not unusual for toughness results and may be caused by the difference in critical conditions for determining their values, *i.e.*, K_{IC} and J_{IC} are determined using the intersections of measured load-displacement curves or R-curves with the secant line with 0.95 slope of initial linear part and 0.2 mm offset line with a slope of $(\sigma_{YS} + \sigma_{UTS})$, respectively.

References for the Response

- [1] Delannay, F., Froyen, L. & Deruyttere, A. The wetting of solids by molten metals and its relation to the preparation of metal-matrix composites. *J Mater. Sci.* **22**, 1-6 (1987).
- [2] Kondoh, K., Kawakami, M., Imai, H., Umeda, J. & Fujii, H. Wettability of pure Ti by molten pure Mg droplets. *Acta Mater.* **58**, 606-614 (2010).
- [3] Hassan, S. F. & Gupta, M. Development of ductile magnesium composite materials using titanium as reinforcement. *J. Alloy. Compd.* **345**, 246-251 (2002).
- [4] Jiang, S., Huang, L. J., An, Q., Geng, L., Wang, X. J. & Wang, S. Study on titanium-magnesium composites with bicontinuous structure fabricated by powder metallurgy and ultrasonic infiltration. *J. Mech. Behav. Biomed. Mater.* **81**, 10-15 (2018).
- [5] Umeda, J., Kawakami, M., Kondoh, K., Ayman, E. S. & Imai H. Microstructural and mechanical properties of titanium particulate reinforced magnesium composite materials. *Mater. Chem. Phys.* **123**, 649-657 (2010).
- [6] Zhang, C. L., Wang, X. J., Wang, X. M., Hu, X. S. & Wu, K. Fabrication,

- microstructure and mechanical properties of Mg matrix composites reinforced by high volume fraction of sphere TC4 particles. *J. Magnes. Alloy.* **4**, 286-294 (2016).
- [7] Zhang, M., Yu, Q., Liu, Z., Zhang, J., Tan, G., Jiao, D., Zhu, W., Li, S., Zhang, Z., Yang, R. & Ritchie, R. O. 3D printed Mg-NiTi interpenetrating-phase composites with high strength, damping capacity, and energy absorption efficiency. *Sci. Adv.* **6**, eaba5581 (2020).
- [8] Zhang, M., Yu, Q., Liu, Z., Zhang, J., Jiao, D., Li, S., Peng, H., Wang, Q., Zhang, Z. & Ritchie, R. O. Compressive properties of 3-D printed Mg-NiTi interpenetrating-phase composite: Effects of strain rate and temperature. *Compos. Part B – Eng.* **215**, 108783 (2021).
- [9] ASTM E915-16, Standard test method for verifying the alignment of X-ray diffraction instrumentation for residual stress measurement, ASTM International, Philadelphia, USA, 2016.
- [10] Zhu, X. K. & Joyce, J. A. Review of fracture toughness (G, K, J, CTOD, CTOA) testing and standardization. *Eng. Fract. Mech.* **85**, 1-46 (2012).
- [11] ASTM E399-90, Standard test method for plane-strain fracture toughness of metallic materials, ASTM International, Philadelphia, USA, 1990.
- [12] ASTM E1820-13, Standard test method for measurement of fracture toughness, ASTM International, Philadelphia, USA, 2013.

REVIEWERS' COMMENTS

Reviewer #1 (Remarks to the Author):

The authors have addressed all my comments/concerns during reading the draft. It's now acceptable for publication.

Reviewer #2 (Remarks to the Author):

Thanks to authors for addressing all comments. This is a very well organized and great piece of research work and deserves to be published. Congratulations!